# The transcription factor Bach2 negatively regulates murine natural killer cell maturation and function

**Shasha Li[1], Michael D Bern[1], Benpeng Miao[2], Changxu Fan[2], Xiaoyun Xing[2], Takeshi Inoue[3], Sytse J Piersma[1], Ting Wang[2], Marco Colonna[4], Tomohiro Kurosaki[3], Wayne M Yokoyama[1]\***

[1]Division of Rheumatology, Department of Medicine, Washington University School of Medicine, St Louis, United States; [2]Department of Genetics, Center for Genome Sciences and Systems Biology, Washington University School of Medicine, St Louis, United States; [3]Laboratory of Lymphocyte Differentiation, WPI Immunology Frontier Research Center, Osaka University, Osaka, Japan; [4]Department of Pathology and Immunology, Washington University School of Medicine, St Louis, United States

**Abstract** BTB domain And CNC Homolog 2 (Bach2) is a transcription repressor that actively participates in T and B lymphocyte development, but it is unknown if Bach2 is also involved in the development of innate immune cells, such as natural killer (NK) cells. Here, we followed the expression of Bach2 during murine NK cell development, finding that it peaked in immature $CD27^+CD11b^+$ cells and decreased upon further maturation. Bach2 showed an organ and tissue-specific expression pattern in NK cells. Bach2 expression positively correlated with the expression of transcription factor TCF1 and negatively correlated with genes encoding NK effector molecules and those involved in the cell cycle. Lack of Bach2 expression caused changes in chromatin accessibility of corresponding genes. In the end, Bach2 deficiency resulted in increased proportions of terminally differentiated NK cells with increased production of granzymes and cytokines. NK cell-mediated control of tumor metastasis was also augmented in the absence of Bach2. Therefore, Bach2 is a key checkpoint protein regulating NK terminal maturation.

**\*For correspondence:**
yokoyama@wustl.edu

## Editor's evaluation

This paper, by identifying the novel factor BACH2s involvement in the generation and maintenance of NK cells, helps to fully define the NK cell network and will contribute to our understanding or regulation of NK cell development and function.

## Introduction

Natural killer (NK) cells are innate lymphoid cells that have spontaneous cytolytic activity against tumor cells and virus-infected cells. The development of NK cells occurs in bone marrow (BM) as well as in secondary lymphoid tissues in both humans and mice. Multipotent hematopoietic stem cells give rise to common lymphoid progenitors (CLPs) that can differentiate into all types of lymphocytes. NK cell precursors (NKP) are then derived and later express IL-2R/IL-15R β chain (CD122), defining refined NK progenitors (rNKp) (*Carotta et al., 2011*; *Fathman et al., 2011*). At this stage in the mouse, commitment to NK cell development occurs, followed by the acquisition of the germline-encoded NK receptors NK1.1 and NKp46, and the cells become immature NK cells. Mature NK cells develop when they gain the expression of DX5 (CD49b), cytotoxic activity, and capacity to produce interferon-γ (IFNγ)

(*Kim et al., 2002*). Mature NK cells can be further defined based upon the differential expression of CD27 and CD11b. Starting from double-negative cells being the most immature cells regarding their functionality, the cells upregulate the expression of CD27 then CD11b to become CD27+CD11b- (CD27+ cells) then CD27+CD11b+ (double-positive) respectively, which undergo homeostatic expansion. Finally, the double-positive cells lose the expression of CD27 and retain expression of CD11b to become terminally differentiated NK cells (CD11b+ cells) with increased cytotoxic activity (*Chiossone et al., 2009*).

The commitment, development, and function of NK cells are distinctly regulated by multiple transcription factors, which are reflected by the expression of many unique surface markers at different stages of NK cell development. Eomesodermin has been identified as a unique factor required for NK development, distinct from ILC1s which share many similarities in surface makers with NK cells (*Gordon et al., 2012*; *Intlekofer et al., 2005*). Kruppel-like factor 2 intrinsically regulates NK cell homeostasis by limiting early-stage NK cell proliferation and guiding them toward trans-presented IL-15 (*Rabacal et al., 2016*). IRF8 is required for the effector functions of NK cells against viral infection (*Adams et al., 2018*). T-bet has a broader function in various cells including T cells, ILC1s, and NK cells, and positively regulates the terminal maturation of NK cells (*Townsend et al., 2004*). Similarly, Zeb2 promotes NK terminal differentiation and may function downstream of T-bet (*van Helden et al., 2015*). Blimp-1 is expressed throughout NK cell maturation and is required for NK homeostasis (*Kallies et al., 2011*). TCF1 (encoded by *Tcf7*) participates in the development of NK cells and its downregulation is required for NK terminal maturation (*Jeevan-Raj et al., 2017*). A multi-tissue single-cell analysis divided NK cells into two major groups based on TCF1 levels: high expression of TCF1 correlated with genes expressed in immature NK cells including *Cd27*, *Xcl1*, and *Kit* while TCF1 was inversely correlated with the expression of genes involved in effector function such as *Gzmb*, *Gzma*, *Ccl5*, and *Klrg1* (*McFarland et al., 2021*). Thus, the expression of unique transcription factors at specific developmental stages of the cells appears to generate distinct gene regulatory circuitries.

These gene regulatory circuits provide 'fingerprints' that may be more reliable to reveal the developmental and functional disparities among phenotypically similar cell populations (*Collins et al., 2019*). Furthermore, they provide insight into how the development of immune cells, including NK cells, occurs. Therefore, it is important to further clarify the expression pattern and function of unique transcription factors for NK cells during their development.

The transcription factor, BTB domain And CNC Homolog 2 (Bach2), was identified as a transcriptional repressor. It forms a heterodimer with Maf family proteins and binds to a DNA motif called T-MARE (TGCTGA G/C TCAGCA), a Maf recognition element (MARE), to regulate gene expression (*Muto et al., 1998*). The regulatory function of Bach2 is mediated through its interaction with the super-enhancers (SEs), and its aberrant expression is associated with a variety of autoimmune diseases as well as cancers (*Afzali et al., 2017*; *Marroquí et al., 2014*; *Roychoudhuri et al., 2016b*). In physiological conditions, Bach2 has been shown to participate in cell development, as previously examined in the adaptive immune system. Bach2 is expressed in CLP and represses genes of myeloid lineage to promote the development of cells in the lymphoid lineage (*Itoh-Nakadai et al., 2014*). Bach2 was first shown to be a B cell-intrinsic transcription factor that regulates B cell development through inhibiting the expression of Blimp-1 (encoded by *Prdm1*) (*Muto et al., 1998*; *Ochiai et al., 2006*). The rapid upregulation of Blimp-1 mediated by Bach2 deficiency promotes the terminal differentiation of B cells toward plasma cells even prior to class-switch recombination (CSR) (*Muto et al., 2004*). Bach2 is also critical in regulating the plasticity of T cells. Under homeostatic conditions, Bach2 maintains T cells in a naïve state, preventing the generation of effector T cells by inhibiting the expression of effector molecules downstream of TCR signaling (*Roychoudhuri et al., 2016a*; *Tsukumo et al., 2013*). Bach2 expression is reduced during T cell polarization while higher expression of Bach2 in CD4+ T cell differentiation promotes the formation of regulatory T cells by repressing genes related to effector differentiation within helper T cell lineages (*Lahmann et al., 2019*; *Roychoudhuri et al., 2016a*; *Roychoudhuri et al., 2013*). Regaining the expression of Bach2 after differentiation downregulated pro-inflammatory signals and differentiation into T cell memory cell lineages (*Herndler-Brandstetter et al., 2018*). Thus, the development and function of the adaptive immune cells are critically controlled by the expression of Bach2 and its associated regulatory circuits.

The role of Bach2 in NK cells has not been characterized. Here, we found that, at steady state, Bach2 was differentially expressed during NK cell development and terminal maturation. Its deficiency

in NK cells resulted in a significantly increased expression of genes involved in NK cytotoxicity. Along with this, NK cells lacking Bach2 expression were more terminally differentiated and demonstrated better control of tumor metastasis. Thus, Bach2 serves as a checkpoint in the terminal maturation of NK cells.

## Results

### Bach2 is expressed at different levels in NK cells at different developmental stages

To characterize the role of Bach2 in NK cell function, we first examined Bach2 expression during different stages of NK development. We used a *Bach2*[Flag] reporter mouse in which a 3xFlag tag was fused at the N-terminus of Bach2 protein as described previously (*Herndler-Brandstetter et al., 2018*). Bach2 expression was detected in CLPs and expression was relatively reduced in pre-NK progenitor cells and rNKp (*Figure 1A* and *Figure 1—figure supplement 1A*), reflecting the critical role of Bach2 in the development of lymphoid cells from CLPs (*Itoh-Nakadai et al., 2014*). During NK specification downstream of CLP, the expression of Bach2 was at a relatively lower level compared to CLPs. At the rNKp stage (CD122⁺CD127⁺), developing NK cells begin to acquire expression of the NK cell markers, NK1.1 and NKp46 (*Fathman et al., 2011*). Bach2 was homogenously expressed by these NK cells in both the BM and spleen (*Figure 1B*). At this stage, NK cells further undergo maturation by acquiring the expression of CD49b (DX5). In both the BM and spleen, Bach2 expression was detected in both immature (DX5⁻NK1.1⁺) and mature (DX5⁺NK1.1⁺) NK cells while mature NK cells had even higher Bach2 expression than the immature NK cells (*Figure 1C* and *Figure 1—figure supplement 1B*). We further subdivided NK cell maturation by surface expression of CD27 and CD11b (*Chiossone et al., 2009*). Surprisingly, the expression of Bach2 was lowest in the most mature CD27⁻CD11b⁺ cells compared with the other two stages (*Figure 1D* and *Figure 1—figure supplement 1B*). This observation was further confirmed by western blot (*Figure 1E*). Thus, during NK cell terminal differentiation, Bach2 may function mainly in the relatively immature stages.

At the bulk splenic NK cell population level, the expression of Bach2 inversely correlated with the age of the mice: 2-week-old mice had the highest expression, and 12-week-old mice had the lowest expression of Bach2 (*Figure 1F*). The overall downregulation of Bach2 with age correlated with changes in the maturation profile of NK cells with age. More than 60% of NK cells were immature CD27⁺CD11b⁻ cells with less than 3% of mature CD27⁻CD11b⁺ cells in 2-week-old mice while around 40% of NK cells in 12-week-old mice were in the most mature CD27⁻CD11b⁺ stage (*Figure 1—figure supplement 1C*). Moreover, 2-week-old NK cells showed higher Bach2 levels than their 12-week-old counterparts in each corresponding NK subset (*Figure 1—figure supplement 1D*). The effect of Bach2 on NK cells may not be restricted to splenic NK cells; we also detected Bach2 expression in NK cells in various different tissues, such as lymph nodes, lung, and liver (*Figure 1G*). Bach2 showed similar expression levels in lymph node and liver as compared to spleen while in the lung, the expression was much lower. In summary, our data showed Bach2 expression in NK cells varies at different developmental stages and shows age and organ-specific patterns.

### Bach2 deficiency drives a transition from immature stem-like phenotype toward mature effector phenotype of NK cells

To further understand the role of Bach2 in NK cell development and function, we generated NK cell-specific Bach2 conditional knockout mice (referred to as Bach2[cKO] mice) by crossing *Bach2*[flox/flox] mice with *Ncr1*[iCre] mice in which *Bach2* was specifically deleted in NKp46-expressing cells. As a control, *Bach2*[flox/flox] mice (referred to as control mice) were used. Examination of Bach2 mRNA showed that exons 1–2, including the start codon, had been deleted in essentially all Bach2[cKO] NK cells (*Figure 2—figure supplement 1A*), confirming deletion of Bach2 expression in Bach2[cKO] NK cells. When we assessed the expression of NK cell inhibitory and activation receptors (Ly49s and NKG2A/CD94), we did not find any changes in the expressed repertoire of these receptors with Bach2 deficiency except CD94 showed significant downregulation (*Figure 2—figure supplement 1B*). However, it was previously reported that genetic deletion of CD94 in NK cells did not affect the development and function of NK cells (*Orr et al., 2010*). Thus, a comprehensive analysis was required to better understand the role of Bach2 in NK cell function.

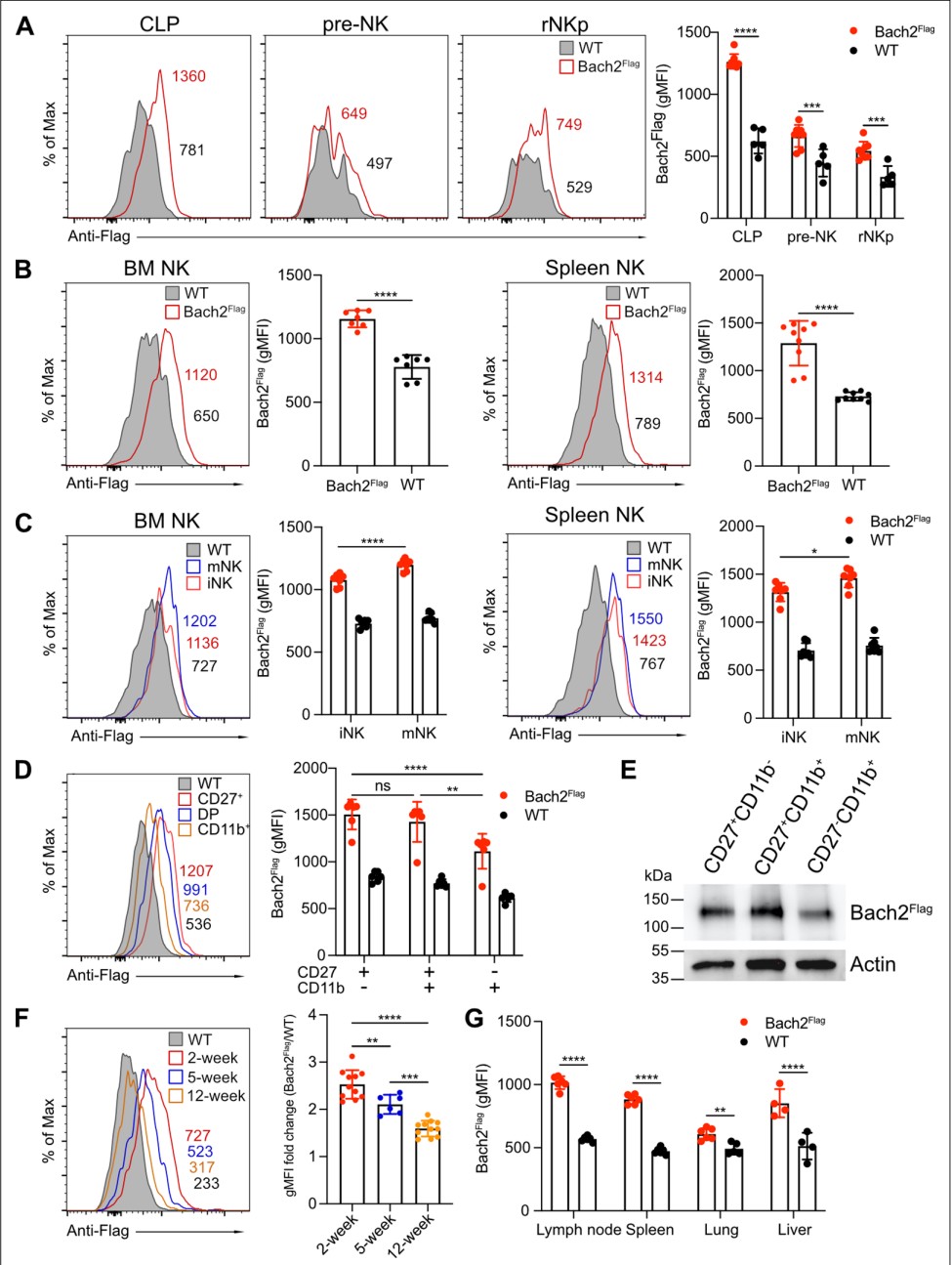

**Figure 1.** BTB domain And CNC Homolog 2 (Bach2) expression at different natural killer (NK) cell developmental stages by analysis of Bach2[Flag] knock-in mouse. (**A**) Histogram plots of Bach2[Flag] expression (red) in common lymphoid progenitor (CLP), pre-NK progenitor (pre-NK), and refined NK progenitor (rNKp) as compared to cells from wild-type (WT) mice (gray fill). The geometric MFI (gMFI) number was indicated in the plot. Summary of gMFI was shown (n=5 for each group in three independent experiments). (**B**) Histogram plots of Bach2[Flag] expression (red) in NK cell (CD3⁻CD19⁻NK1.1⁺) from bone marrow (BM) and spleen as compared to cells from WT mice (gray fill). Numbers indicate gMFI and summary of gMFI is shown (n=7 or 9 for each group in three to four independent experiments). (**C**) CD3⁻CD122⁺ NK cells from BM and spleen were subdivided into iNK (DX5⁻NK1.1⁺) and mNK (DX5⁺NK1.1⁺) and analyzed for Bach2[Flag] expression (red for iNK, blue for mNK) as compared to cells from WT mice (gray fill). Numbers indicate gMFI and summary of gMFI is shown (n=7 for each group in three to four independent experiments). (**D**) Histogram plot of Bach2[Flag] expression in CD27⁺CD11b⁻ (red), CD27⁺CD11b⁺ (blue), and CD27⁺CD11b⁺ (yellow) NK subsets as compared to cells from WT mice (gray fill). Note that WT plot represents expression of Bach2[Flag] from CD27⁻CD11b⁺ subsets. Numbers indicate gMFI and summary of gMFI is shown (n=6 for each group in three independent experiments). (**E**) Splenic NK cells were sorted into CD27⁺CD11b⁻,

*Figure 1 continued on next page*

*Figure 1 continued*

CD27⁺CD11b⁺ and CD27⁻CD11b⁺ subsets. Bach2^Flag expression in the subsets was detected using Anti-FLAG M2-Peroxidase (HRP) antibody by western blot. Expression of Actin was used as an internal control. Two individual experiments have been done with one mouse each time. (**F**) Histogram plot of Bach2^Flag expression in splenic NK cell (CD3⁻CD19⁻NK1.1⁺) from mice at 2 weeks' (red), 5 weeks' (blue), and 12 weeks' (orange) age as compared to cells from WT mice (gray fill). Note that WT plot represents expression of Bach2^Flag from 12-week age mice. Numbers indicate gMFI and summary of gMFI fold change is shown (n=6 or 11 for each group in three to four independent experiments). (**G**) Summary of gMFI data for Bach2^Flag expression in NK cells from lymph node, spleen, lung, and liver (n=4 or 6 for each group in three independent experiments). NK cells from lymph node, spleen, and lung were gated as CD3⁻CD19⁻NK1.1⁺. NK cells from liver were gated as CD3⁻CD19⁻NK1.1⁺DX5⁺. Statistical significance was determined by two-way ANOVA (**A, C, D, and G**), one-way ANOVA, (**F**) or by Student's t test (**B**). Error bars indicate SD. *p<0.05; **p<0.01; ***p<0.001; ****p<0.0001. ns, not significant. See *Figure 1— figure supplement 1* for gating strategies.

The online version of this article includes the following source data and figure supplement(s) for figure 1:

**Source data 1.** Bach2 expression in different subsets by western blot.

**Figure supplement 1.** Gating strategy for flow cytometry analysis.

---

To this end, we performed RNA-seq analysis compared between Bach2-deficient NK cells and Bach2-sufficient NK cells using sorted splenic NK cells (*Figure 2—figure supplement 1C*). Principal component analysis (PCA) revealed significant changes transcriptionally in Bach2-deficient NK cells as compared to control NK cells (*Figure 2—figure supplement 1D*). Overall, we found 133 genes downregulated and 210 genes upregulated in Bach2^cKO NK cells as compared to control NK cells (Bach2^cKO versus control) (*Figure 2A*). The transcripts with decreased expression corresponded to genes involved in T cell differentiation, cell development, and cell homeostasis pathways (*Figure 2A*). These gene signatures suggested that Bach2 controls NK cell differentiation. Specifically, the top 2 downregulated genes were *Kit* and *Tcf7* (*Figure 2B*), which were previously shown to be responsible in maintaining the stemness of the T cells (*Siddiqui et al., 2019*). Regarding NK cells, the loss of *Tcf7* expression led to enhanced NK cell terminal maturation (*Jeevan-Raj et al., 2017*). We also detected the downregulation of *Cd27*, *Ccr7*, and *Cd69* (*Figure 2B*). The downregulation of *Ccr7* was previously shown to be correlated with human NK cell differentiation toward effector phenotype (*Hong et al., 2012*). On the other hand, the transcripts with elevated expression included genes involved in the cell cycle, cell proliferation, and inflammatory response pathways (*Figure 2A*), suggesting a skewing toward an effector phenotype as a result of Bach2 deficiency. Indeed, among the upregulated genes were those associated with NK cell effector functions such as *Klrg1*, *Gzmb*, *Gzmk*, and *Ccl5* (*Figure 2B*; *Bezman et al., 2012*), indicating that lack of Bach2 expression facilitated the differentiation of NK cells toward terminal maturation.

We confirmed the transcriptomics data by quantitative PCR (qPCR) analysis performed on sorted splenic NK cells from the Bach2^cKO and control mice (*Figure 2C and D*). We detected significantly lower expression of genes, for example, *Tcf7*, *Kit*, and *Sox6* in Bach2-deficient NK cells. For other genes such as *Cd69*, *Sell*, and *Ccr7*, we did not observe significant differences, but they all had a trend of downregulation in NK cells lacking Bach2 expression (*Figure 2C*). Regarding genes upregulated in the RNA-seq analysis, we confirmed *Cd39*, *Klrg1*, *Gzmb*, *Klrb1b*, and *Ccl5* to be increased by qPCR analysis. However, *Cx3cr1* did not reach significance even though it displayed a trend toward upregulation (*Figure 2D*). Consistent with their transcription level, the proteins encoded by these genes were changed accordingly with Bach2 deficiency. Kit, CD69, and TCF1 (encoded by *Tcf7*) were significantly downregulated in Bach2-deficient NK cells while CD62L (encoded by *Sell*) did not show difference at the protein level (*Figure 2E and F*). On the other hand, CD39, KLRG1, CX3CR1, and Granzyme B were elevated, consistent with their changes at the RNA level (*Figure 2E and F*). Although we detected increased expression of multiple genes involved in cell cycle and proliferation, we did not find a difference in Ki67 expression between Bach2^cKO and control NK cells (*Figure 2—figure supplement 1E*). Nonetheless, we generally confirmed that our RNA-seq data were reliable to reflect the impact of Bach2 in regulating the expression of various genes in NK cells.

Next, we asked how Bach2 participated in NK cell biology. It was reported that the enforced Bach2 expression in exhausted CD8⁺ T cells resulted in the cells becoming exclusively stem-like precursor, exhausted CD8⁺ T cells, preventing their further differentiation into terminal exhausted CD8⁺ T cells

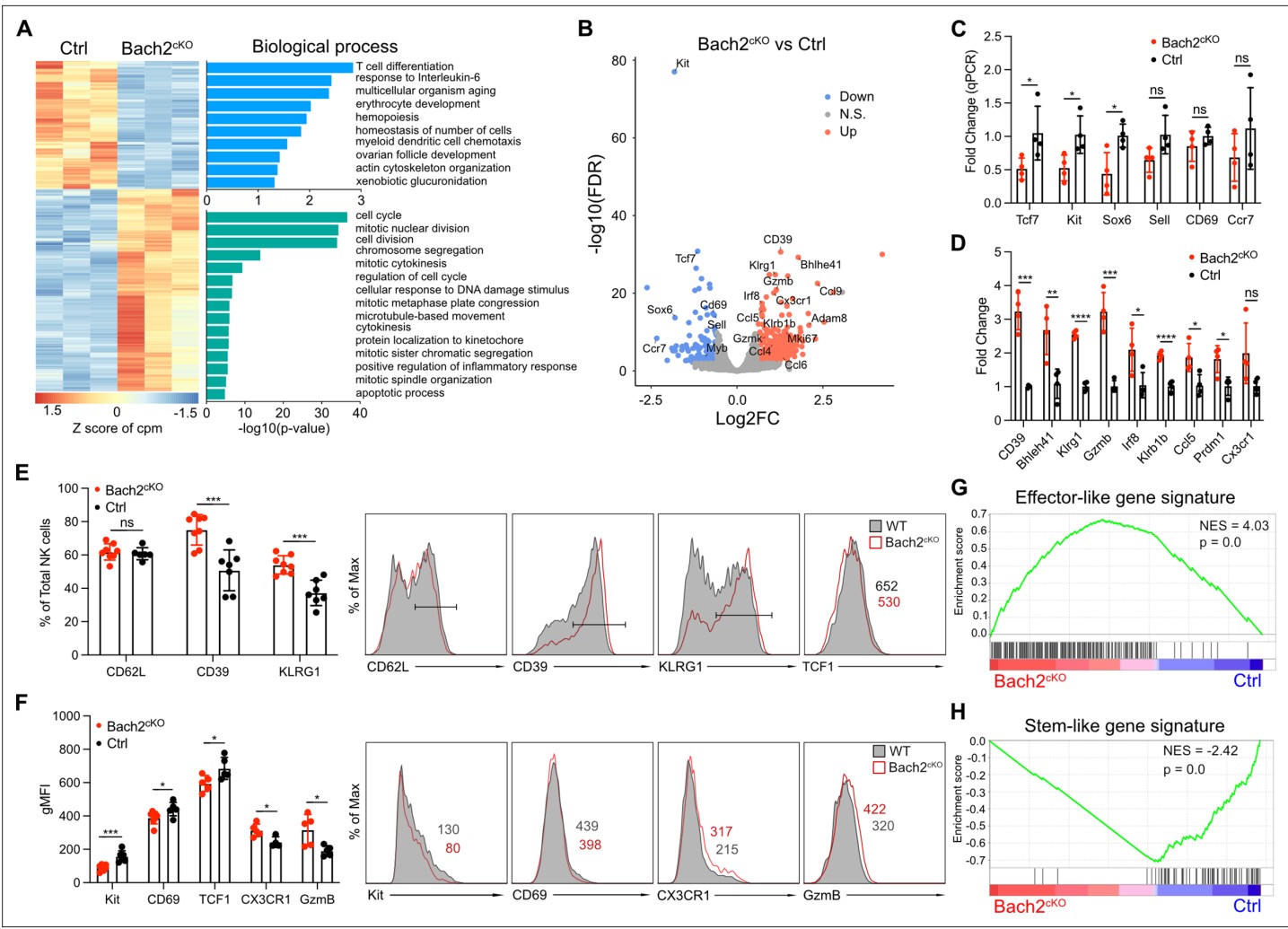

**Figure 2.** RNA-seq analysis reveals BTB domain And CNC Homolog 2 (Bach2) deficiency in natural killer (NK) cells promotes the terminal maturation of NK cells with elevated effector function. (**A**) Heatmap of differentially expressed genes in NK cells compared between control and Bach2cKO mice in RNA-seq analysis (FDR < 0.01 and log2 fold change >1.5). Each column represents total splenic CD3-CD19-NK1.1+NKp46+ cells from an individual mouse. Three biological replicates per group from two individual sorting experiments are shown. The data were analyzed with the Database for Annotation, Visualization and Integrated Discovery (DAVID) Gene Ontology (GO) analysis for the biological process using the genes differentially expressed from NK cells between control and Bach2cKO mice. (**B**) Volcano plot shows differential gene expression between control and Bach2cKO splenic NK cells. Highlighted are genes discussed in the text. (**C and D**) RNA expression of indicated genes in Bach2cKO mice or control mice determined by quantitative PCR (qPCR). Data are shown with four mice per group from two independent experiments. (**E**) Summary of the percentage of NK cells expressing the indicated genes in Bach2cKO mice or control mice (n=7 or 8 for each group in three independent experiments). Representative histogram plots show the expression of indicated protein in NK cells from Bach2cKO mice (red) as compared to NK cells from control mice (gray fill). (**F**) Summary of the geometric MFI (gMFI) of indicated gene in total NK cells in Bach2cKO mice or control mice (n=5–8 for each group in three independent experiments). Representative histogram plots show the expression of indicated proteins in NK cells in Bach2cKO mice (red) as compared to NK cells from control mice (gray fill). (**G and H**) Gene set enrichment analysis (GSEA) illustrating enrichment of effector-like (**G**) and stem-like (**H**) gene signatures in Bach2cKO and control splenic NK cells. Statistical significance was determined by Student's t test. Error bars indicate SD. *p<0.05; **p<0.01; ***p<0.001; ****p<0.0001. ns, not significant.

The online version of this article includes the following figure supplement(s) for figure 2:

**Figure supplement 1.** BTB domain And CNC Homolog 2 (Bach2) deficiency in natural killer (NK) cells resemble activated effector CD8+ T cells.

(*Yao et al., 2021*). We used gene set enrichment analysis (GSEA) to determine the effect of Bach2 deficiency in NK cells by comparison to the gene signatures of stem-like CD8+ T cells and terminally differentiated effector-like CD8+ T cells (*Utzschneider et al., 2016*). We found genes upregulated in NK cells induced by Bach2 deficiency positively correlated with terminal differentiated effector-like gene signatures (*Figure 2G*) whereas the genes downregulated showed a stem-like signature (*Figure 2H*).

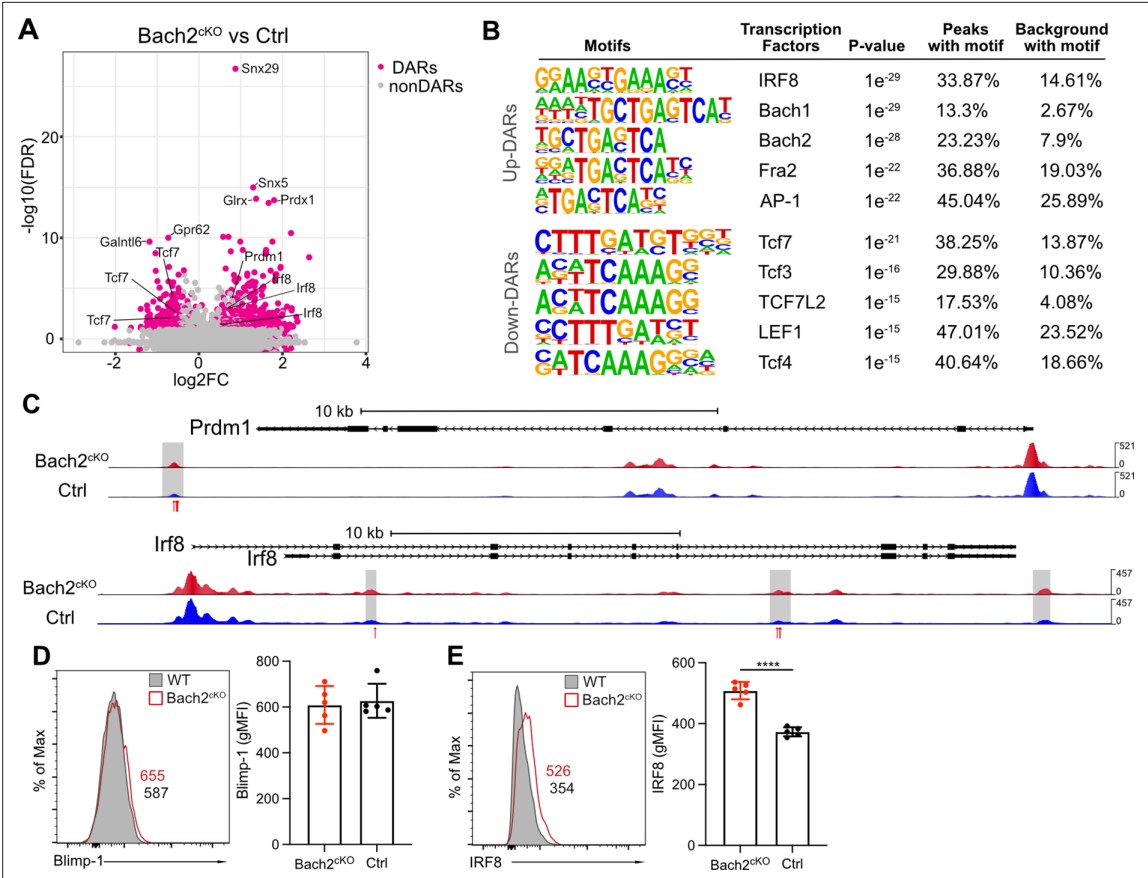

**Figure 3.** Bach2 deficiency results in increased accessible regions in the genome of natural killer (NK) cells. (**A**) Volcano plot shows the differentially accessible regions (DARs) and non-DARs between control and Bach2cKO splenic NK cells. (**B**) De novo motif enrichment at regions of open chromatin as defined by assay for transposase-accessible chromatin coupled with high throughput sequencing (ATAC-seq) in splenic NK cells from Bach2cKO mice compared to cells from control mice. Motifs of interest are listed. (**C**) Genome browser visualization of ATAC-seq peak near *Prdm1* and *Irf8* gene loci in NK cells from Bach2cKO mice (red) and control mice (blue). Gray boxes denote differential accessible regions. Red arrows show the location of Bach2-binding motif. (**D and E**) Histogram plot shows the expression of Blimp-1 (**D**) and IRF8 (**E**) in NK cells from Bach2cKO mice (red) as compared to control mice (gray fill). Numbers indicate geometric MFI (gMFI). Summary of gMFI is shown (n=5 for each group in three independent experiments). Statistical significance was determined by Student's t test. Error bars indicate SD. *p<0.05; ****p<0.0001.

The online version of this article includes the following figure supplement(s) for figure 3:

**Figure supplement 1.** Potential protein interactions between BTB domain And CNC Homolog 2 (Bach2) and differentially regulated transcription factors in natural killer (NK) cells.

The increased effector gene signatures induced by Bach2 deficiency in NK cells were further validated by comparing our RNA-seq data with the differential gene expression data generated from naïve and effector CD8[+] T cells (*Roychoudhuri et al., 2016a*). We found that Bach2-sufficient NK cells displayed a naïve CD8[+] T cell signature (*Figure 2—figure supplement 1F*) while Bach2-deficient NK cells resembled activated effector CD8[+] T cells (*Figure 2—figure supplement 1G*). These data suggested that Bach2 expression suppressed terminal differentiation of NK cells by repressing many effector genes.

## Bach2 deficiency in NK cells results in altered chromatin accessibility

To understand the mechanisms that drive these unique gene expression programs regulated by Bach2, we performed assay for transposase-accessible chromatin coupled with high throughput sequencing (ATAC-seq) to identify active or poised regulatory elements in isolated NK cells from Bach2cKO and control mice at a genome-wide level. Within the differentially accessible regions (DARs) identified by ATAC-seq, 564 regions were found to be upregulated and 251 were found to be downregulated in Bach2-deficient NK cells as compared to Bach2-sufficient NK cells (*Figure 3A*). Thus, with Bach2

deficiency, the DARs indicated overall increased chromatin accessibility in the genome, consistent with the putative function of Bach2 as a repressor in NK cells.

The upregulated DARs contained motifs bound by the transcription factors IRF8, Bach1, Bach2, and AP-1 (*Figure 3B*). Prior studies indicated IRF8-mediated signaling is required for NK cell maturation and function (*Mace et al., 2017*), suggesting increased accessibility of IRF8-regulated genes in Bach2 deficiency thus may result in bias toward NK cell development. Interestingly, we found that the enriched binding motifs by Bach1, Bach2, Fra2 (Fos-related Antigen 2, an AP-1 transcription factor subunit), and AP-1 in upregulated DARs all contained a common binding sequence (5′-TGA(C/G)TCA-3′) (*Figure 3B*). In CD8$^+$ T cells, Bach2 has been reported to share this common binding sequence with the AP-1 transcription factor family and compete with AP-1 factors for the access to the target genes due to the similarity in their binding motifs (*Roychoudhuri et al., 2016a*). Indeed, we found the direct target genes of Bach2 in CD8$^+$ T cells, such as *Irf8*, *Ccl3*, *Ccl4*, and *Gzmb*, showed elevated expression in Bach2$^{cKO}$ NK cells (*Figure 3—figure supplement 1A*). Also, there was increased JunD occupancy in the absence of Bach2 (*Figure 3—figure supplement 1B*) in NK cells. Thus, these data suggest that Bach2 in NK cells may also compete for AP-1-binding sites to regulate target genes and that the deficiency of Bach2 may increase accessibility for AP-1 transcription factors (such as JunD) to occupy the shared motif.

The motifs enriched in downregulated DARs were mostly TCF/LEF family protein-binding motifs (*Figure 3B*), especially the binding motif for TCF1 (encoded by *Tcf7*) that was also one of the top downregulated genes in Bach2$^{cKO}$ NK cells (*Figure 2E and F*). Deficiency of *Tcf7* and its downstream genes promoted the terminal differentiation of NK cells as well as CD8$^+$ T cells (*Jeevan-Raj et al., 2017*; *Utzschneider et al., 2016*). Thus, the reduced accessibility of *Tcf7* targeted motifs in the absence of Bach2 reinforced our findings that Bach2 acted as a checkpoint protein to maintain the stemness of NK cells.

As a transcription repressor, Bach2 functions through interacting with other transcription factors or repressing the expression of downstream target genes. For example, Blimp-1 has been widely studied as a direct target of Bach2 in B cell and T cells (*Kometani et al., 2013*; *Muto et al., 2004*; *Ochiai et al., 2006*; *Roychoudhuri et al., 2013*). To explore the transcription factors affected by Bach2 deficiency, we analyzed the differentially expressed genes in RNA-seq and lowered the threshold (FDR < 0.05 and log2 fold change > 0.2) to include more transcription factors (*Figure 3—figure supplement 1C*). Most of these differentially expressed transcription factors formed an intricate interaction network linked by Bach2 shown by the STRING database (*Figure 3—figure supplement 1D*). Bach2 closely interacted with several transcription factors in the list such as Blimp-1 (encoded by *Prdm1*), TCF1 (encoded by *Tcf7*), BCL6, IRF8, and Fosl2 (*Figure 3—figure supplement 1D*). Since Bach2 functions as transcription repressor in NK cells, the upregulation of the transcripts of *Prdm1* and *Irf8* in Bach2-deficient NK cells indicates they may be targets of Bach2 in wild-type (WT) NK cells (*Figure 3—figure supplement 1C*). By ATAC-seq we found that both *Prdm1* and *Irf8* genes displayed increased accessibility with Bach2 deficiency (*Figure 3C*). Furthermore, there were Bach2-binding motifs in the DARs, suggesting a direct regulatory function of Bach2 on *Prdm1* and *Irf8* (*Figure 3C*). In Bach2-deficient NK cells, *Prdm1* and *Irf8* were shown to be upregulated at RNA level by both RNA-seq and qPCR (*Figure 2D*). However, at the protein level, we failed to detect the corresponding changes of Blimp-1 compared between NK cells from Bach2$^{cKO}$ and control mice (*Figure 3D*). While this may reflect differences in kinetics of mRNA versus protein expression, Blimp-1 has been shown to be a target of Bach2 in development of functional B cells and T cells (*Kometani et al., 2013*; *Muto et al., 2004*; *Ochiai et al., 2006*; *Roychoudhuri et al., 2013*), suggesting another possibility that the regulatory circuit may be different between Bach2 and Blimp-1 in NK cells. Surprisingly, we found that IRF8 was significantly upregulated in Bach2-deficient NK cells (*Figure 3E*). Given that the loss of IRF8 resulted in impaired terminal maturation of NK cells (*Mace et al., 2017*), Bach2 may repress the expression of IRF8 directly or indirectly to affect NK cell development. Overall, our data showed that Bach2 deficiency increased accessibility across the genome in NK cells and Bach2 may directly compete for AP-1-binding sites as a repressor or function in an indirect manner through interaction with other transcription factors.

## Bach2 restrained terminal maturation of NK cells

Our genomics and transcriptomics analyses, coupled with the expression pattern of Bach2 during NK cell development, suggested that Bach2 restricted the terminal maturation of immature NK cells.

To address this, we analyzed the maturation profile of the NK cells in Bach2$^{cKO}$ and control mice. In the BM, we did not observe any significant difference between control and Bach2$^{cKO}$ mice regarding the frequency of the CD27$^+$ cells or CD11b$^+$ cells (*Figure 4A*) although total NK cell numbers were reduced in Bach2$^{cKO}$ mice (*Figure 4B*). We detected an altered maturation profile of NK cells in the spleen of Bach2$^{cKO}$ mice with more cells having the mature NK cell phenotype (CD11b$^+$) and fewer cells at the immature DP stage as compared to control mice (*Figure 4C*). However, the total number of NK cells was unchanged in the spleen (*Figure 4D*). As we previously showed, the expression of KLRG1, a marker of terminally mature splenic NK cells was also increased at the population level (*Figure 2E*), further confirming biased development into terminally mature NK cells. To confirm the lack of influence of T cells and B cells on NK cell development and maturation, we evaluated NK cells in germline Bach2-deficient mice on the *Rag1$^{-/-}$* background. The maturation of NK cells was analyzed in both *Bach2$^{-/-}$ Rag1$^{-/-}$* (referred to as Bach2$^{-/-}$) mice and *Rag1$^{-/-}$* (referred to as WT) mice. Similar to the results presented in Bach2$^{cKO}$ mice, we found the frequency of CD27$^+$CD11b$^+$ NK cells was significantly reduced while the frequency of CD27$^-$CD11b$^+$ NK cells was markedly increased in both BM (*Figure 4—figure supplement 1A*) and spleen (*Figure 4—figure supplement 1B*) in Bach2$^{-/-}$ mice. In agreement, the expression of KLRG1 was upregulated and we also found more cells (~80%) expressing KLRG1 in Bach2$^{-/-}$ mice compared to Bach2-sufficient mice (~50%) (*Figure 4—figure supplement 1C*). We also examined NK maturation profiles in other tissues. We did not find significant changes in NK cell maturation in lymph nodes (*Figure 4E*) whereas a reduced NK cell number was found (*Figure 4F*). In liver, we observed a similar pattern as we found in spleen, with increased terminal maturation of NK cells (*Figure 4G*) but we failed to see changes in NK cell number (*Figure 4H*). Interestingly, lung displayed a unique profile of NK maturation with most of the NK cells falling into the most mature NK subset in control mice (*Figure 4I*). Even while biased toward this subset, we still detected an increased percentage of CD27$^-$CD11b$^+$ cells with Bach2 deficiency (*Figure 4I*). Different from other tissues, we also detected more NK cells in lung from Bach2$^{cKO}$ mice (*Figure 4J*), suggesting a preferential homing of terminally matured NK cells into the lung. Taken together, these data indicate that NK cells were skewed toward the most mature subset in Bach2-deficient mice as compared to Bach2-sufficient mice, with a concomitant decrease in immature NK cells.

## B16 tumor growth and metastasis is controlled by NK cells with Bach2 deficiency

Since Bach2-deficient NK cells showed skewed differentiation with increased expression of effector molecules, especially cytotoxic genes and increased cell numbers in lung, we evaluated whether mice lacking Bach2 in NK cells would result in better control of tumor lung metastases in vivo. We injected 2.5×10$^5$ B16F10 cells intravenously into Bach2$^{cKO}$ or control mice and evaluated tumor metastases 2 weeks later by counting the black colonies formed in lungs. The lung metastases of B16F10 tumors in Bach2$^{cKO}$ mice were dramatically reduced compared to control mice (*Figure 5A and B*). This control of tumor metastases was NK cell-dependent because both Bach2$^{cKO}$ mice and control mice displayed a similar number of metastatic tumor colonies when NK cells were depleted with anti-NK1.1 (PK136) (*Figure 5A and B*). These data suggested that Bach2 deficiency plays a role in controlling tumor metastasis by NK cells.

To determine if Bach2 deficiency in NK cells resulted in better direct killing of B16F10 cells, we performed in vitro tumor killing assays with B16F10 cells. However, we did not see any difference in killing B16F10 cells between Bach2$^{cKO}$ and control NK cells (*Figure 5—figure supplement 1A*). We also assessed IFNγ production in response to various stimulations including anti-NK1.1, IL-15, and IL-12 plus IL-18 on splenic NK cells. We did not detect any difference with or without Bach2 expression in NK cells (*Figure 5—figure supplement 1B*). Although splenic NK cells lacking Bach2 showed increased Granzyme B expression, lung NK cells displayed similar levels of Granzyme B expression between control and Bach2$^{cKO}$ mice at steady state (*Figure 5—figure supplement 1C*). Since it had been shown that NK cells effectively cleared disseminated tumor cells from the lung within 24 hr of tumor inoculation (*Ichise et al., 2022*), we tested the NK function in vivo in the lung 24 hr after tumor injection in Bach2$^{cKO}$ mice. We observed increased NK cells in the lung in Bach2$^{cKO}$ mice (*Figure 5C*) similar to the phenotype at steady state (*Figure 4J*). We did not detect more proliferating NK cells in the lungs in Bach2$^{cKO}$ mice after tumor injection (*Figure 5—figure supplement 1D*), suggesting that NK cell migration may cause the increased lung NK cell numbers in Bach2$^{cKO}$ mice. NK cells also

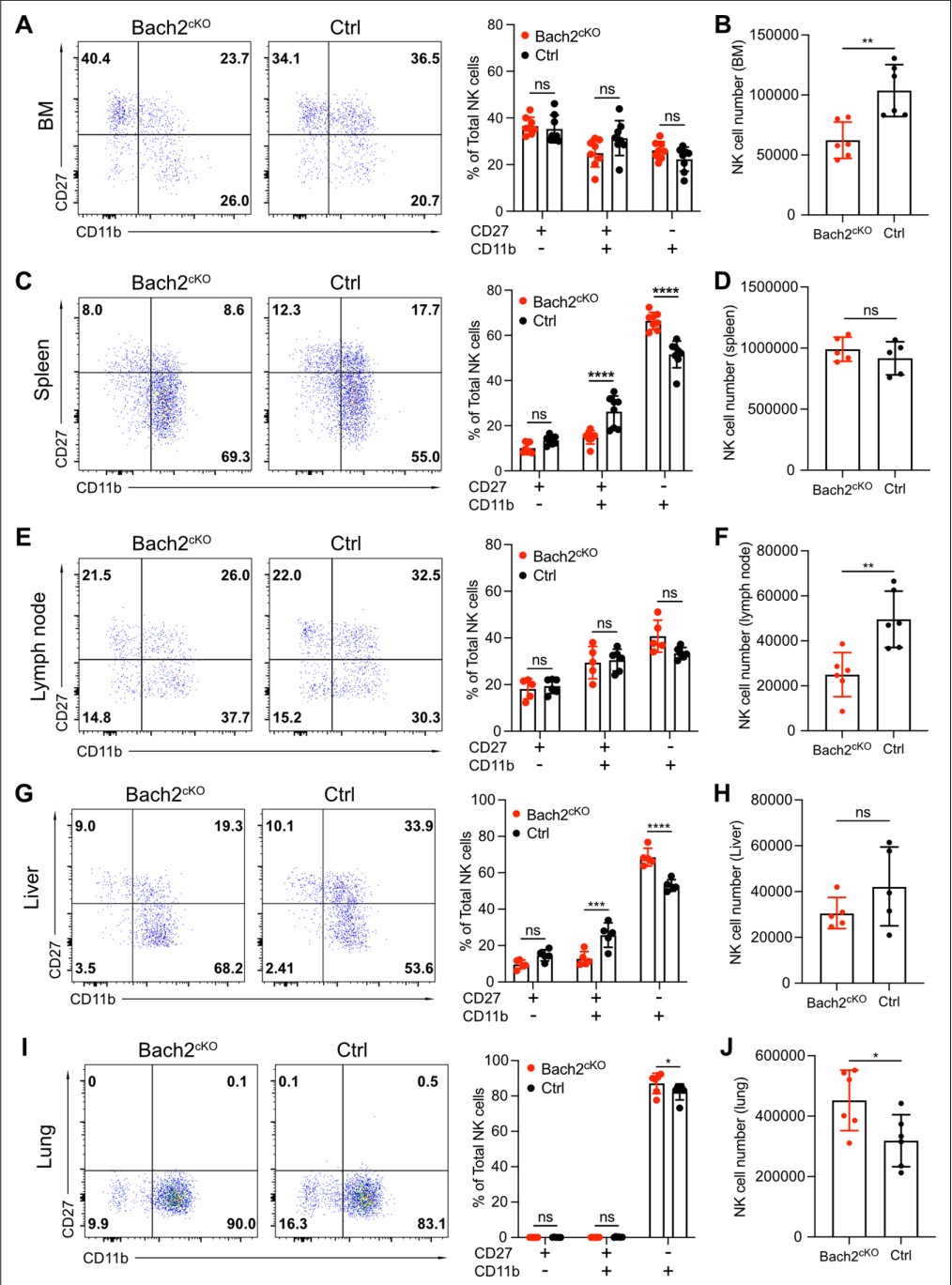

**Figure 4.** Bach2 deficiency increases natural killer (NK) cells with terminally differentiated phenotype. Representative flow cytometry plots show NK cells (CD3⁻CD19⁻NK1.1⁺NKp46⁺) separated into maturation stages by CD27 and CD11b expression. The percentage of CD27⁺CD11b⁻, CD27⁺CD11b⁺, CD27⁻CD11b⁺ NK cells among all NK cells (left, representative flow plots; right, bar graph summary) from bone marrow (BM) (**A**), spleen (**C**), lymph node (**E**), liver, (**G**) and lung (**I**) in Bach2^cKO mice and control mice were plotted (n=5–8 for each group from three independent experiments). Note that NK cells from liver were gated on CD3⁻CD19⁻DX5⁺NK1.1⁺NKp46⁺. Summary of the total number of NK cells from BM (**B**), spleen (**D**), lymph node (**F**), liver (**H**), and lung (**J**) in Bach2^cKO mice and control mice were plotted (n=5–8 for each group from three independent experiments). Statistical significance was determined by two-way ANOVA (**A, C, E, G, and I**) or by Student's t test (**B, D, F, H, and J**). Error bars indicate SD. *p<0.05; **p<0.01; ***p<0.001; ****p<0.0001. ns, not significant.

The online version of this article includes the following figure supplement(s) for figure 4:

**Figure supplement 1.** BTB domain And CNC Homolog 2 (Bach2) deficiency results in a more mature phenotype.

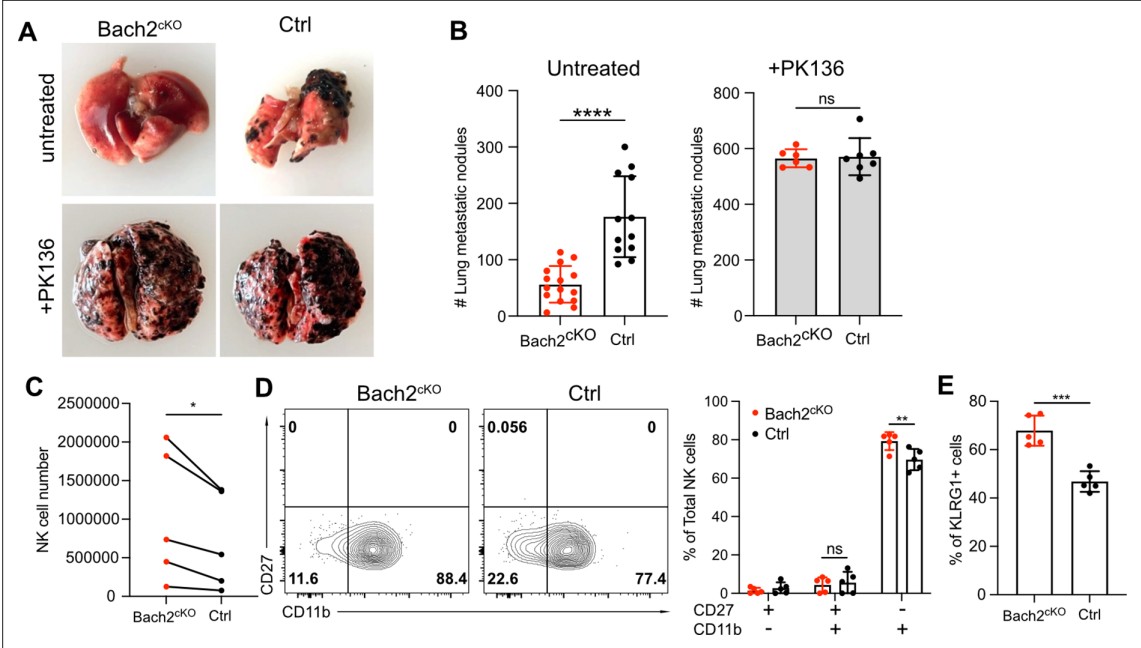

**Figure 5.** Lack of BTB domain And CNC Homolog 2 (Bach2) expression in natural killer (NK) cells suppresses B16F10 tumor metastasis.
(**A**) Representative picture of lung metastatic nodules in Bach2cKO mice and control mice under steady-state or anti-NK1.1 (PK136) depletion.
(**B**) Summary of the number of B16F10 metastatic nodules in lung from Bach2cKO and control mice with or without anti-NK1.1 (PK136) depletion. Data were pooled from three to four independent experiments with a total of six to fifteen mice per group. (C–E) $2 \times 10^5$ B16F10 cells were intravenously injected to Bach2cKO mice or control mice. After 24 hr, NK cells from lung were harvested and analyzed for total number (**C**), maturation by CD27 and CD11b (**D**), and the percentage of KLRG1+ NK cells (**E**). Average NK cell numbers were calculated for each experiment with one to two mice per group, and data were pooled together from five independent experiments (**C**). Data were pooled from three independent experiments with five mice per group (**D and E**). Statistical significance was determined by Student's t test (**B and E**), Student's t test (paired t test) (**C**), or by two-way ANOVA (**D**). Error bars indicate SD. *p<0.05; **p<0.01; ***p<0.001; ****p<0.0001. ns, not significant.

The online version of this article includes the following figure supplement(s) for figure 5:

**Figure supplement 1.** Natural killer (NK) cell effector function was unchanged with BTB domain And CNC Homolog 2 (Bach2) deficiency.

displayed a biased maturation profile toward CD27-CD11b+ NK cells (*Figure 5D*) as well as increased proportion of KLRG1+ cells (*Figure 5E*) in Bach2cKO mice compared to control mice after tumor challenge. Studies have shown that the terminally developed CD27lowKLRG1+CD11bhigh NK subsets are crucial in clearing the early metastatic colorectal carcinoma in mice (*Malaisé et al., 2014*). Thus, the increased NK cell number with higher percentage of mature CD27-CD11b+ subset with elevated KLRG1 expression may be responsible for the lower metastatic colonies in the lung in Bach2cKO mice. In summary, NK cells in Bach2-deficient mice are more efficient in controlling tumor progression and metastasis in the lung.

## Discussion

Here, we found that Bach2 is highly expressed in functional immature NK cells (CD27+ cells) and gradually downregulates its expression at the terminal stage of NK maturation (CD11b+ cells). In line with this, we demonstrated that Bach2 deficiency caused a biased NK cell development toward terminal differentiation in an NK cell-intrinsic manner. Bach2-deficient NK cells also displayed increased cytotoxic gene expression and chromatin accessibility and were more potent in controlling tumor metastases.

Bach2 plays a critical role in the development of lymphoid cells: its presence in CLP repressed the expression of genes important for myeloid cells, promoting the development of T and B cells (*Itoh-Nakadai et al., 2014*). However, in the BM, the expression of Bach2 increased after commitment of stem cells to the B cell lineage with its expression high in pre-pro B cells, pro-B cells, pre-B cells, and immature B cells. Bach2 served as a checkpoint protein in B cells that inhibited the expression

of the immunoglobulin heavy chain of activated p53 by competing with BCL6 for functional VDJ rearrangements (*Muto et al., 1998*; *Swaminathan et al., 2013*). Bach2 also suppressed the differentiation of activated B cells to plasma cells by inhibiting the expression of Blimp-1 (encoded by *Prdm1* gene), which allowed CSR and somatic hypermutation to take place before becoming plasma cells (*Kometani et al., 2013*; *Muto et al., 2004*; *Ochiai et al., 2006*). Here, we also detected a relatively high expression of Bach2 in CLP but there was a reduction in pre-NK progenitors and refined NK progenitors. We showed that Bach2 started to gain its expression in NK cells after the acquisition of germline-encoded NK receptors such as NK1.1 and NKp46, a stage when NK cells displayed an effector program. The differential expression pattern of Bach2 in the early stage between NK cell development and B cell development indicated a divergent trajectory of CLP for the commitment of B cells and NK cells.

Bach2 was specifically highly expressed in CD27$^+$ cells but not in CD11b$^+$ terminal differentiated cells. Interestingly, maturing NK cells upregulate CD27 transiently followed by the upregulation of CD11b and KLRG1 (*Chiossone et al., 2009*; *Huntington et al., 2007*). During this transition, NK cells lose their homeostatic expansion capacity but acquire cytotoxic activity (*Chiossone et al., 2009*; *Huntington et al., 2007*), suggesting the main role of Bach2 is to suppress the functional development of the most mature NK cells. Indeed, when we examined the differential gene expression and genome accessibility in the context of Bach2 deficiency specifically in mature NK cells, we detected an upregulation of a series of genes related to cell proliferation, immune effector molecules, and cell apoptosis. In contrast, genes associated with cell development and homeostasis were downregulated with Bach2 deficiency, suggesting that Bach2 might be required for NK cell self-renewal at steady state.

The Bach2-regulated genes we detected with differential expression patterns in NK cells correlated very well with the gene signatures observed in CD8$^+$ T cells, their cytotoxic counterparts in the adaptive immune system. One of the genes, *Tcf7*, was particularly interesting and may be important for the mechanism of the regulation of NK development by Bach2. *Tcf7* (encoding TCF1) is highly expressed in naïve T cells, decreased in effector T cells, and regained its expression in memory cells, showing its role in maintaining pluripotency of the T cells (*Willinger et al., 2006*; *Zhao et al., 2010*). Similarly, Bach2 also maintained T cells in a naïve status under homeostatic conditions, preventing the generation of effector T cells through inhibiting the expression of effector molecules downstream of TCR signaling (*Roychoudhuri et al., 2016a*; *Tsukumo et al., 2013*). On the other hand, TCF1 was recently shown to be a hallmark of stem-like precursor exhausted CD8$^+$ T cells with self-renewal capability and can differentiate into terminal effector-like exhausted CD8$^+$ T cells which lacked TCF1 expression (*Utzschneider et al., 2020*). It was shown that Bach2 also played a positive role in maintaining the pool of these stem-like precursors exhausted CD8$^+$ T cells as enforced overexpression of Bach2 resulted in the cells retaining this stem-like condition while knockout of Bach2 led to terminal differentiation of the cells (*Yao et al., 2021*). Our data in NK cells also showed the link between Bach2 and TCF1 as Bach2 deficiency caused downregulation of TCF1 transcription. More importantly, TCF1 has been previously shown to participate in NK development and its downregulation was required for NK cell terminal maturation (*Jeevan-Raj et al., 2017*), suggesting that Bach2 and TCF1 may interact in regulating NK cell development.

Given that Bach2 is a transcriptional repressor through its interaction with SEs and TCF1 was downregulated in the absence of Bach2, it is possible that Bach2 may indirectly regulate TCF1 expression. Although RNA-seq, ATAC-seq, and flow cytometry all showed a trend for the downregulation of TCF1, it is still unknown how TCF1 is regulated by Bach2. Unlike CD8$^+$ T cells in which Bach2 may antagonize RUNX3 to induce TCF1 expression (*Yao et al., 2021*), we did not detect changes in RUNX3 expression in NK cells, indicating a different regulatory circuit between Bach2 and TCF1 in NK cells. Another potential interaction between Bach2 and TCF1 is through BCL6. We found that BCL6 was downregulated in Bach2$^{cKO}$ NK cells while BCL6 was recently reported to directly bind the regulatory region of *Tcf7* and promoted its expression in memory CD8$^+$ T cells (*Liu et al., 2019*), indicating that Bach2 can induce BCL6 expression (a similar positive correlation was found in B cells) (*Huang et al., 2014*) and promote TCF1 expression. Thus, the pathways for Bach2 to regulate TCF1 may incorporate pathways shared by both B cells and T cells but since we only detected a minor decrease of TCF1 at the protein level in Bach2-deficient NK cells, to what extent Bach2-TCF1 regulation impacts NK development requires further investigation.

Regarding how Bach2 functions in NK cells as compared to other lymphoid cells, Blimp-1 was uncovered to be the target of Bach2 in both B cells and T cells to regulate cell differentiation and function (*Ochiai et al., 2006*; *Roychoudhuri et al., 2013*). Regulome analysis of human NK cells also proposed that Bach2-mediated gene suppression relied upon inhibiting Blimp-1 expression, and Blimp-1 repressed the TCF1-LEF1-MYC-induced homeostatic expansion of NK cells (*Collins et al., 2019*). However, in our data, we did not detect changes in Blimp-1 protein expression except for a slight upregulation at the mRNA level in Bach2-deficient NK cells, indicating *Prdm1* is not the major target of Bach2 in regulating NK cell differentiation. Interestingly, IRF8 may be potentially regulated by Bach2 to promote NK cell maturation. IRF8 expression was significantly upregulated in Bach2-deficient NK cells and the chromatin at the *Irf8* locus showed increased accessibility. Bach2 may repress IRF8 transcription directly since we found Bach2-binding motifs within the intron region. On the other hand, Bach2 may interact with other transcription factors to regulate IRF8 expression indirectly. A recent study shows that there are three MafK-binding sites within the *Irf8* locus (*Fourier et al., 2020*). As Bach2 has been shown to act together with MafK to bind to the MAREs and negatively regulate B cell function (*Muto et al., 1998*), it is possible that a similar mechanism may occur in NK cell regulation by Bach2.

It is of great interest to explore other gene regulatory circuits that control NK cell developmental stages. While Bach2 negatively regulates downstream factors to restrain NK terminal differentiation, the expression and function of Bach2 are expected to be tightly regulated by upstream transcription factors. Studies in B cells showed that the mammalian target of rapamycin complex 2 (mTORC2) inhibits FoxO1 to reduce Bach2 mRNA expression (*Tamahara et al., 2017*). Interestingly, FoxO1 was found to directly bind within and proximal to the Bach2 gene locus in CD8 T cells (*Delpoux et al., 2021*). Upon T cell activation, FoxO1 induces the Bach2 transcripts to compete with AP-1 factors for AP-1-binding sites (*Delpoux et al., 2021*). In NK cells, FoxO1 suppresses NK cell maturation by repressing T-bet (*Deng et al., 2015*) and mTORC2 was also found to regulate T-bet expression through Akt$^{S473}$- FoxO1 (*Yang et al., 2018*). When mTORC2 was knocked out, the expression of T-bet was impaired. However, Bach2 was upregulated at the mRNA level as a target of FoxO1 (*Yang et al., 2018*). Together, these data suggest that Bach2 may be the downstream target of an mTORC2-AKT-FoxO1 axis in NK cells, which will need further evaluation.

Human NK cells encompass two major subsets, known as CD56$^{dim}$ and CD56$^{bright}$ NK cells. CD62L (encoded by *Sell*) and CCR7 were shown to be highly expressed by CD56$^{bright}$ NK cells and drove their migration to secondary lymphoid tissues (*Campbell et al., 2001*; *Frey et al., 1998*). CD56$^{bright}$ NK cells also expressed high levels of c-Kit for their homeostatic proliferation (*Matos et al., 1993*). In agreement with this, *Sell*, *Ccr7*, and *Kit* genes were all downregulated in Bach2-deficient NK cells in our data in mice. In contrast, CD56$^{dim}$ NK cells displayed a high density of CX3CR1 for the migration to tissues and higher cytotoxic activity by increased expression of perforin and various granzymes (*Campbell et al., 2001*), which resembled our Bach2-deficient NK phenotypes in mice. More importantly, Bach2 has been demonstrated to be highly expressed by CD56$^{bright}$ NK cells and with low expression in CD56$^{dim}$ NK cells (*Holmes et al., 2021*). In murine NK cells, Bach2 also shows preferential expression in immature CD27$^+$ subsets compared to mature CD11b$^+$ subsets. Therefore, the function of Bach2 may be conserved between human and mouse for its regulatory circuitries but this still needs further exploration.

NK cells are currently being studied in clinical trials as potential targets for cancer immunotherapy. Our study shows that, in mice, Bach2 functions as a checkpoint to restrain NK cell cytotoxicity and Bach2 deficiency leads to enhanced NK cell-mediated control of B16 melanoma metastases. Studies in humans have also suggested that Bach2 may play a similar role in human NK cells (*Collins et al., 2019*). As a result, our study suggests that Bach2 may be a novel target for checkpoint inhibition of NK cells for cancer immunotherapy.

## Materials and methods
### Mice and cell line

WT C57BL/6 (B6) mice and *Rag1*$^{-/-}$ mice were purchased from The Jackson Laboratories. *Bach2*$^{Flag}$ knock-in mice have been described before (*Herndler-Brandstetter et al., 2018*) as have *Bach2*$^{flox/flox}$ mice (*Kometani et al., 2013*). NK cell Bach2 conditional knockout mice were generated by crossing

*Bach2*<sup>flox/flox</sup> mice with *Ncr1*<sup>iCre</sup> mice (**Narni-Mancinelli et al., 2011**) from Eric Vivier (CNRS-INSERM-Universite de la Mediterranee, Marseille, France). ES cells for *Bach2*<sup>-/-</sup> (*Bach2*<sup>tm1e</sup>) mice were purchased from the EuComm program. The mice were derived from ES clone EPD0689_1_H03, ES line JM8A3. N1. Animal experiments were performed with 6- to 12-week male or female mice, except for those specifically indicated. The B16F10 cell line was bought from ATCC (Catalog# CRL-6475). The identity was confirmed by ATCC and there is no mycoplasma contamination.

## Antibodies and flow cytometry

The following antibodies and reagents were purchased from eBioscience (San Diego, CA): anti-CD127 (A7R34), anti-CD3e (145–2C11), anti-CD19 (eBio1D3), anti-CD49b (DX5), anti-NK1.1 (clone PK136), anti-CD27 (LG.TF9), anti-CD11b (M1/70), anti-NKp46 (29A1.4), anti-CD39 (24DMS1), anti-Granzyme B (NGZB), anti-IFNγ (XMG1.2), anti-Ly49A (eBio12A8), anti-Ly49D (eBio4E5), anti-Ly49E/F (CM4), anti-Ly49I (YLI-90), anti-CD94 (18d3), anti-NKG2A (16a11),anti-TER-119 (TER-119), anti-CD62L (MEL-14), anti-CD117 (2B8), anti-IRF8 (V3GYWH), anti-Blimp-1 (5E7), anti-Ki-67 (SolA15), Fixable Viability Dye eFluor 506. The following antibodies were purchased from BD Biosciences (San Diego, CA): anti-CD244.2 (2B4), anti-TCF-7/TCF-1 (S33-966), anti-Ly49F (HBF-719), anti-CD122 (TM-β1), anti-CD69 (H1.2F3), anti-Ly-49G2 (4D11). The following antibodies were purchased from Biolegend (San Diego, CA): anti-CD135 (A2F10.1), anti-DYKDDDDK tag (L5), anti-KLRG1 (MAFA) (2F1/KLRG1), anti-CX3CR1 (SA011F11). Anti-Ly49H (3D10) and anti-Ly49C (4LO33) were produced in-house. BM or spleen cells were treated with RBC lysis buffer to remove erythrocytes. Then, cells were treated with 2.4G2 (anti-Fc RII/III) hybridoma supernatants (produced in-house) to block Fc receptors. Surface staining was performed on ice in FACS staining buffer (1% BSA, 0.01% NaN₃ in PBS). The labeling of BM progenitor populations has been described before (**Jeevan-Raj et al., 2017**). Lineage-positive cells were labeled by a cocktail of biotin-conjugated anti-CD3e, CD19, NK1.1, CD11b, Gr-1, Ter-119 antibodies (eBioscience). The resulting lineage-negative cells (Lin⁻) were further stained to identify CLP, pre-NK progenitor, and rNK progenitor as indicated. For intracellular staining of Bach2<sup>Flag</sup>, TCF1, Blimp-1, and IRF8, the Foxp3 transcription factor staining buffer set (BD Biosciences) was used according to the manufacturer's protocols. For intracellular staining of GzmB and IFNγ, the intracellular fixation and permeabilization buffer set (BD Biosciences) according to the manufacturer's protocols, samples were collected by FACS Canto (BD Biosciences), and data were analyzed by FlowJo.

## Western blot

NK cells from spleen were enriched with the EasySep Mouse NK cell isolation kit (STEMCELL Technologies, Vancouver, BC) according to the manufacturer's instructions. Enriched NK cells were then labeled by indicated surfaced markers and sorted into different subsets. Sorted cells were lysed in RIPA buffer in the presence of Halt Protease Inhibitor Cocktail (Thermo Fisher Scientific, Catalog# 78429, Waltham, MA) on ice. Lysates were denatured in ×2 Laemmli sample buffer (Bio-Rad, Hercules, CA) and resolved by SDS-PAGE (Bio-Rad, 4561034). Proteins were transferred to nitrocellulose membranes and probed with indicated antibodies. Anti-FLAG M2-Peroxidase (HRP) (A8592) was purchased from Sigma (St Louis, MO). Beta-Actin Rabbit antibody (4967S) and anti-rabbit IgG HRP-linked antibody (7074S) were purchased from Cell Signaling (Danvers, MA).

## RNA-seq and qPCR

NK cells from the spleen were enriched by the EasySep Mouse NK cell isolation kit (STEMCELL Technologies) according to the manufacturer's instructions. Enriched cells were then sorted into CD3⁻CD19⁻NK1.1⁺NKp46⁺ cells. RNA was purified by PureLink RNA Mini Kit (Invitrogen, 12183018A). RNA-seq was performed by the Genome Technology Access Center at Washington University School of Medicine. NovaSeq 6000 was used for sequencing. RNA-seq reads were then aligned to the Ensembl release 76 primary assembly with STAR v2.5.1a (**Dobin et al., 2013**). Gene counts were derived from the number of uniquely aligned unambiguous reads by Subread/featureCount v1.4.6-p5 (**Liao et al., 2014**). Low expressing genes were filtered with the criteria of cmp > 1 in at least three samples. Two outliers were removed. RUVr method (k=1) in RUVseq R package was used to remove batch effect. Differential gene expression was determined using the EdgeR R package with FDR < 0.01 and log2 fold change > 1.5 as the thresholds. Heatmaps were generated with pheatmap R package. PCA was performed by prcomp function of R. Gene set enrichment pathways analysis was done using the

Broad Institute's GSEA software by comparing signature databases from GSE83978 and GSE77857. For qPCR, cDNA was synthesized by ProtoScript II Reverse Transcriptase (NEB, M0368S, Ipswich). Pre-designed primers for indicated genes were obtained from IDT. Quantitative real-time PCR was performed by PowerUp SYBR Green Master Mix Kit (Thermo Fisher Scientific) on a StepOnePlus real-time PCR system (Thermo Fisher Scientific). Relative gene expression was normalized to beta-actin and calculated by the ΔΔCt method.

## ATAC-seq and analyses

NK cells from the spleen were enriched by the EasySep Mouse NK cell isolation kit (STEMCELL Technologies) according to the manufacturer's instructions. Enriched cells were then sorted into CD3$^-$CD19$^-$NK1.1$^+$NKp46$^+$ cells. ATAC-seq libraries were generated using the omni-ATAC protocol (*Corces et al., 2017*); 150K NK cells were used as input for each sample. All ATAC libraries were barcoded, pooled, and sequenced using 75 bp paired-end read to a depth of at least 30 million. Quality control of ATAC data was performed using the AIAP pipeline (v1.1) (*Liu et al., 2021*). All libraries received the highest possible score (total score = 10). ATAC peaks for each sample were called as a part of the AIAP pipeline, using macs2 with FDR cutoff at 0.01. After manual inspection of the peaks called, we further filtered the resulting narrowPeak files for peaks with −log10FDR≥9. To call DARs, we merged all overlapping peaks to create a union peak set. We then formed a peak-sample matrix with each entry corresponding to the number of Tn5 insertion sites in each peak for each sample. Tn5 insertion sites were also calculated as a part of the AIAP pipeline. DARs were called using DESeq2 (v1.26.0) (*Love et al., 2014*) with default parameters with the cutoff of 'log2 fold change≥0.5 and FDR≤0.05'. For motif enrichment in DARs, we separately generated background peaks for up (Bach2$^{cKO}$ > Control) and down (Control > Bach2$^{cKO}$) DARs. For each DAR, we selected without replacement its 10 closest non-DARs in the Euclidean space constructed from WT samples as background peaks. We used HOMER (v4.11.1) (*Heinz et al., 2010*) for motif enrichment in DARs relative to background peaks (findMotifsGenome.pl up_DAR.bed mm10.fa out_dir -bg up_DAR_background.bed -nomotif -size given). To annotate Bach2 motifs in DARs, we downloaded the Bach2 motif (MA1101.1) from the JASPAR (*Castro-Mondragon et al., 2022*) motif database, and used FIMO (v5.4.1) (*Grant et al., 2011*) to scan DARs for the presence of this motif with a p-value cutoff at 0.0005. ATAC signal was visualized using WashU Epigenome Browser (*Li et al., 2022*).

## B16F10 metastasis assay

B16F10 cells were maintained in R10 (RPMI 1640 medium [Gibco] containing 10% FBS, 1% penicillin/streptomycin, 1% L-glutamine, 55 µM 2-mercaptoethanol). Before injection, $2.5×10^5$ B16F10 cells were resuspended in 300 µl PBS, and intravenously injected into mice. After 14 days, tumor metastasis was evaluated by counting the black colonies formed in the lung under a dissecting microscope with a blinded analysis whereby the investigator counting the nodules was unaware of the experimental conditions.

## Cell isolation and stimulation

Single-cell suspensions were obtained from spleen, BM, mesenteric lymph node, liver, and lung as previously described (*Sojka et al., 2014*). Additionally, in order to obtain cells from lung, the tissue was subjected to collagenase (Liberase Blendzyme III, Roche), hyaluronidase (Sigma), and DNase I (grade II, Roche) for digestion (*Wu et al., 2015*). Splenocytes were stimulated with 1 µg/ml precoated antibody against NK1.1 (PK136), 100 ng/ml IL-15 (PeproTech), or 20 ng/ml IL-12 p70 (PeproTech) plus 20 ng/ml IL-18 (MBL, Woburn, MA). BFA (Biolegend) was added after 1 hr. After 6 hr of stimulation, cells were harvested for IFNγ detection by flow cytometry.

## NK cell cytotoxicity assay

Lymphokine-activated killer (LAK) cells were produced by passing splenocytes through nylon wool column then cultured in R10 with 800 U/ml IL-2 supernatant (homemade) for 7 days. Adherent LAK cells from day 4, day 6, and day 7 were checked for the purity by flow cytometry for CD3$^-$CD19$^-$NK1.1$^+$ cells. B16F10 cells were labeled by 40 µM calcein-AM (Biolegend) as target cells (*Neri et al., 2001*). LAK cells were then co-cultured with target cells at different effector to target ratios for 4 hr. Target cells alone were treated with 2% Triton for maximal release. Cell-free supernatants were harvested and

fluorescence (excitation: 488 nm; emission: 520 nm) determined on Synergy H1 plate reader (Center for Human Immunology and Immunotherapy Programs, Washington University School of Medicine). The specific killing efficiency was calculated as {(experimental release – spontaneous release)/(maximal release – spontaneous release)} * 100.

## Statistical analysis

Statistical analyses were performed using GraphPad Prism 9 software. The statistical test used is stated in the figure legends. Data are presented as mean ± SD as stated in the figure legends. Statistical significance was determined as indicated with $p < 0.05$ was considered statistically significant.

## Acknowledgements

We thank Susan Gilfillan for helping with the establishment of the Bach2^{-/-} (Bach2^{tm1e}) mice. We thank Beatrice Plougastel-Douglas and Anna Sliz for helpful discussions. This work was supported by NIH grants R01-AI129545 (WMY), R01HG007175 (TW), U01HG009391 (TW), and U41HG010972 (TW).

## Additional information

### Competing interests

Tomohiro Kurosaki: Reviewing editor, *eLife*. The other authors declare that no competing interests exist.

### Funding

| Funder | Grant reference number | Author |
|---|---|---|
| National Institute of Allergy and Infectious Diseases | R01-AI129545 | Wayne M Yokoyama |
| National Human Genome Research Institute | R01-HG007175 | Ting Wang |
| National Human Genome Research Institute | U01-HG009391 | Ting Wang |
| National Human Genome Research Institute | U41-HG010972 | Ting Wang |

The funders had no role in study design, data collection and interpretation, or the decision to submit the work for publication.

### Author contributions

Shasha Li, Conceptualization, Data curation, Formal analysis, Investigation, Visualization, Methodology, Writing - original draft, Project administration, Writing - review and editing; Michael D Bern, Conceptualization, Data curation, Formal analysis, Investigation, Visualization, Methodology; Benpeng Miao, Formal analysis, Investigation, Visualization, Methodology; Changxu Fan, Data curation, Formal analysis; Xiaoyun Xing, Formal analysis, Methodology; Takeshi Inoue, Marco Colonna, Tomohiro Kurosaki, Resources, Methodology; Sytse J Piersma, Data curation, Formal analysis, Methodology; Ting Wang, Formal analysis, Supervision, Funding acquisition, Project administration; Wayne M Yokoyama, Conceptualization, Resources, Software, Supervision, Funding acquisition, Investigation, Writing - original draft, Project administration, Writing - review and editing

### Author ORCIDs

Shasha Li ![ORCID] http://orcid.org/0000-0001-5924-6396
Sytse J Piersma ![ORCID] http://orcid.org/0000-0002-5379-3556
Marco Colonna ![ORCID] http://orcid.org/0000-0001-5222-4987
Tomohiro Kurosaki ![ORCID] http://orcid.org/0000-0002-6352-304X
Wayne M Yokoyama ![ORCID] http://orcid.org/0000-0002-0566-7264

### Ethics

Mouse studies were conducted in accordance with the institutional ethical guidelines through institutional animal care and use committee (IACUC) protocol that was approved by the Animal Studies Committee of Washington University (#20180293).

### Decision letter and Author response

Decision letter https://doi.org/10.7554/eLife.77294.sa1
Author response https://doi.org/10.7554/eLife.77294.sa2

## Additional files

### Supplementary files

• Transparent reporting form

### Data availability

RNA-sequencing data have been deposited in NCBI's Gene Expression Omnibus and are accessible through GEO Series accession number GSE196530. ATAC-seq data have been deposited in NCBI's Gene Expression Omnibus and are accessible through GEO Series accession number GSE212807. Previously published datasets are available on NCBI's Gene Expression Omnibus under the accession number GSE83978 and GSE77857.

The following datasets were generated:

| Author(s) | Year | Dataset title | Dataset URL | Database and Identifier |
|---|---|---|---|---|
| Li S, Miao B, Yokoyama WM | 2022 | Bach2 negatively regulates natural killer cell maturation and effector program | https://www.ncbi.nlm.nih.gov/geo/query/acc.cgi?acc=GSE196530 | NCBI Gene Expression Omnibus, GSE196530 |
| Li S, Fan C, Xing X, Wang T, Yokoyama WM | 2022 | Chromatin accessibility of transcription factor Bach2 in mouse NK cells (ATAC-seq) | https://www.ncbi.nlm.nih.gov/geo/query/acc.cgi?acc=GSE212807 | NCBI Gene Expression Omnibus, GSE212807 |

The following previously published datasets were used:

| Author(s) | Year | Dataset title | Dataset URL | Database and Identifier |
|---|---|---|---|---|
| Charmoy M, Pais Ferreira D, Calderon-Copete S, Chennupati V, Pradervand S, Held W | 2016 | Memory-like CD8 T-cells sustain the immune response to chronic viral infections | https://www.ncbi.nlm.nih.gov/geo/query/acc.cgi?acc=GSE83978 | NCBI Gene Expression Omnibus, GSE83978 |
| Roychoudhuri R, Clever D, Li P, Wakabayashi Y, Quinn KM, Klebanoff CA, Ji Y, Sukumar M, Eil RL, Yu Z, Spolski R, Palmer DC, Pan JH, Patel SJ, Macallan DC, Fabozzi G, Shih H, Kanno Y, Muto A, Zhu J, Gattinoni L, O'Shea JJ, Okkenhaug K, Igarashi K, Leonard WJ, Restifo NP | 2016 | BACH2 regulates CD8+ T cell differentiation by controlling access of AP-1 factors to enhancers | https://www.ncbi.nlm.nih.gov/geo/query/acc.cgi?acc=GSE77857 | NCBI Gene Expression Omnibus, GSE77857 |

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

# Appendix 1

## Appendix 1—key resources table

| Reagent type (species) or resource | Designation | Source or reference | Identifiers | Additional information |
|---|---|---|---|---|
| Strain, strain background (*Mus musculus*) | C56BL/6J, wild type | The Jackson Laboratories | RRID:IMSR_JAX:000664 | |
| Strain, strain background (*Mus musculus*) | *RAG1*−/−, wild type | The Jackson Laboratories | RRID:IMSR_JAX:034159 | |
| Strain, strain background (*Mus musculus*) | *Bach2*Flag | Tomohiro Kurosaki | | |
| Strain, strain background (*Mus musculus*) | *Bach2*flox/flox, flox-Bach2 | Tomohiro Kurosaki | | |
| Strain, strain background (*Mus musculus*) | NKp46iCre; *Ncr1*tm1-1(icre)Viv | Eric Vivier | MGI:5308410 | |
| Strain, strain background (*Mus musculus*) | *Bach2*−/−, *Bach2*tm1e | EuComm | Bach2tm1e(EUCOMM)Wtsi | ES clone EPD0689_1_H03, ES line JM8A3.N |
| Cell line (*Mus musculus*) | B16F10 | ATCC | CRL-6475 | |
| Antibody | Anti-Mouse CD127 PE-Cyanine7, rat monoclonal | eBioscience | Catalog# 25-1271-82 | Flow cytometry (1:100) |
| Antibody | Anti-Mouse CD3e FITC, armenian hamster monoclonal | eBioscience | Catalog# 11-0031-85 | Flow cytometry (1:100) |
| Antibody | Anti-Mouse CD19 FITC, rat monoclonal | eBioscience | Catalog# 11-0193-86 | Flow cytometry (1:100) |
| Antibody | Anti-Mouse CD49b (Integrin α2) FITC, rat monoclonal | eBioscience | Catalog# 11-5971-85 | Flow cytometry (1:100) |
| Antibody | Anti-Mouse NK1.1 PE-Cyanine7, mouse monoclonal | eBioscience | Catalog# 25-5941-82 | Flow cytometry (1:100) |
| Antibody | Anti-Mouse CD27 APC, armenian hamster monoclonal | eBioscience | Catalog# 17-0271-82 | Flow cytometry (1:100) |
| Antibody | Anti-Mouse CD11b eFluor 450, rat monoclonal | eBioscience | Catalog# 48-0112-82 | Flow cytometry (1:100) |
| Antibody | Anti-Mouse NKp46 PerCP-eFluor 710, rat monoclonal | eBioscience | Catalog# 46-3351-82 | Flow cytometry (1:100) |
| Antibody | Anti-Mouse CD39 PE, rat monoclonal | eBioscience | Catalog# 12-0391-82 | Flow cytometry (1:100) |
| Antibody | Anti-Mouse Granzyme B PE, rat monoclonal | eBioscience | Catalog# 12-8898-80 | Flow cytometry (1:100) |
| Antibody | Anti-Mouse IFNγ gamma eFluo450, rat monoclonal | eBioscience | Catalog# 48-7311-82 | Flow cytometry (1:100) |
| Antibody | Anti-Mouse Ly49A/D PE, rat monoclonal | eBioscience | Catalog# 12-5783-81 | Flow cytometry (1:100) |
| Antibody | Anti-Mouse Ly49D APC, rat monoclonal | eBioscience | Catalog# 17-5782-82 | Flow cytometry (1:100) |
| Antibody | Anti-Mouse Ly49E/F APC, rat monoclonal | eBioscience | Catalog# 17-5848-80 | Flow cytometry (1:100) |
| Antibody | Anti-Mouse Ly49I FITC, mouse monoclonal | eBioscience | Catalog# 11-5895-82 | Flow cytometry (1:100) |
| Antibody | Anti-Mouse CD94 FITC, rat monoclonal | eBioscience | Catalog# 11-0941-82 | Flow cytometry (1:100) |
| Antibody | Anti-Mouse NKG2A PerCP eFluor 710, mouse monoclonal | eBioscience | Catalog# 46-5897-82 | Flow cytometry (1:100) |
| Antibody | Anti-Mouse TER-119 Biotin, rat monoclonal | eBioscience | Catalog# 13-5921-85 | Flow cytometry (1:100) |
| Antibody | Anti-Mouse CD117 APC, rat monoclonal | eBioscience | Catalog# 17-1171-81 | Flow cytometry (1:100) |
| Antibody | Anti-Mouse CD62L eFluor450, rat monoclonal | eBioscience | Catalog# 48-0621-82 | Flow cytometry (1:100) |

*Appendix 1 Continued on next page*

*Appendix 1 Continued*

| Reagent type (species) or resource | Designation | Source or reference | Identifiers | Additional information |
|---|---|---|---|---|
| Antibody | Anti-Mouse CD122 eFluor 450, rat monoclonal | eBioscience | Catalog# 48-1222-82 | Flow cytometry (1:100) |
| Antibody | Anti-Mouse CD244.2 FITC, mouse monoclonal | BD Pharmingen | Catalog# 553305 | Flow cytometry (1:100) |
| Antibody | Anti-Mouse TCF-7/TCF-1 PE, mouse monoclonal | BD Pharmingen | Catalog# 564217 | Flow cytometry (1:100) |
| Antibody | Anti-Mouse Ly49F PE, mouse monoclonal | BD Pharmingen | Catalog# 550987 | Flow cytometry (1:100) |
| Antibody | Anti-Mouse CD122 FITC, rat monoclonal | BD Pharmingen | Catalog# 553361 | Flow cytometry (1:100) |
| Antibody | Anti-Mouse CD69 PE, armenian hamster monoclonal | BD Pharmingen | Catalog# 553237 | Flow cytometry (1:100) |
| Antibody | Anti-Mouse Ly49-G2 APC, rat monoclonal | BD Pharmingen | Catalog# 555316 | Flow cytometry (1:100) |
| Antibody | Anti-Mouse CD135 Brilliant Violet 421, rat monoclonal | Biolegend | Catalog# 135313 | Flow cytometry (1:100) |
| Antibody | Anti-DYKDDDDK tag APC, rat monoclonal | Biolegend | Catalog# 637307 | Flow cytometry (1:100) |
| Antibody | Anti- DYKDDDDK tag PE, rat monoclonal | Biolegend | Catalog# 637310 | Flow cytometry (1:100) |
| Antibody | Anti-Mouse KLRG1 (MAFA) Biotin, syrian hamster monoclonal | Biolegend | Catalog# 138406 | Flow cytometry (1:100) |
| Antibody | Anti-Mouse CX3CR1 PE, mouse monoclonal | Biolegend | Catalog# 149005 | Flow cytometry (1:100) |
| Antibody | Anti-Mouse Ly49H Biotin, mouse monoclonal | In-house | | Flow cytometry (1:100) |
| Antibody | Anti-Mouse Ly49C Biotin, mouse monoclonal | In-house | | Flow cytometry (1:100) |
| Antibody | Anti-Mouse CD3 Biotin, armenian hamster monoclonal | eBioscience | Catalog# 145–2C11 | Flow cytometry (1:100) |
| Antibody | Anti-Mouse CD19 Biotin, rat monoclonal | eBioscience | Catalog# 13-0193-85 | Flow cytometry (1:100) |
| Antibody | Anti-Mouse NK1.1 Biotin, mouse monoclonal | Biolegend | Catalog# 108704 | Flow cytometry (1:100) |
| Antibody | Anti-Mouse CD11b Biotin, rat monoclonal | Biolegend | Catalog# 101204 | Flow cytometry (1:100) |
| Antibody | Anti-Mouse TER-119 Biotin, rat monoclonal | eBioscience | Catalog# 13-5921-85 | Flow cytometry (1:100) |
| Antibody | Anti-Mouse Ly-6G and Ly-6C Biotin, rat monoclonal | BD Pharmingen | Catalog# 553125 | Flow cytometry (1:100) |
| Antibody | Anti-Mouse CD135 Biotin, rat monoclonal | BD Horizon | Catalog# 562537 | Flow cytometry (1:100) |
| Antibody | Anti-Human/Mouse IRF8 APC, mouse monoclonal | eBioscience | Catalog# 17-9852-82 | Flow cytometry (1:100) |
| Antibody | Anti-Mouse Blimp-1 PE, rat monoclonal | eBioscience | Catalog# 12-9850-82 | Flow cytometry (1:100) |
| Antibody | Anti-Mouse Ki-67 eFluor, rat monoclonal | eBioscience | Catalog# 48-5698-82 | Flow cytometry (1:100) |
| Peptide, recombinant protein | Streptavidin PerCP | BD Pharmingen | Catalog# 554064 | Flow cytometry (1:200) |
| Peptide, recombinant protein | Streptavidin PE | BD Pharmingen | Catalog# 554061 | Flow cytometry (1:200) |
| Other | Fixable Viability Dye eFluor 506 | eBioscience | Catalog# 65-0866-14 | Flow cytometry (1:1000) |
| Antibody | Anti-FLAG M2-Peroxidase (HRP) mouse monoclonal | Sigma | Catalog# A8592 | Western blot (1:1000) |

*Appendix 1 Continued on next page*

*Appendix 1 Continued*

| Reagent type (species) or resource | Designation | Source or reference | Identifiers | Additional information |
|---|---|---|---|---|
| Antibody | Beta-Actin antibody rabbit polyclonal | Cell Signaling | Catalog# 4967S | Western blot (1:1000) |
| Antibody | Anti-rabbit IgG HRP-linked antibody affinity purified from goat | Cell Signaling | Catalog# 7074S | Western blot (1:1000) |
| Commercial assay or kit | EasySep Mouse NK cell isolation kit | STEMCELL Technologies | Catalog# 19855 | NK cell enrichment |
| Commercial assay or kit | PureLink RNA Mini Kit | Ambion | Catalog# 12183018A | RNA isolation |
| Sequence-based reagent | Tcf7 | IDT DNA | Mm.PT.58.13528979 | Fwd: CTTCAATCTGCTCATGCCCTA Rev: TGTTCGTAGAGTGGAGAAAGC |
| Sequence-based reagent | Gzmb | IDT DNA | Mm.PT.58.42155916 | Fwd: AAGAGAGCAAGGACAACACTC Rev: CATGTCCCCCGATGATCTC |
| Sequence-based reagent | Klrg1 | IDT DNA | Mm.PT.58.30803964 | Fwd: GCTCACATCTCCTTACATTTCCG Rev: TCCTCAAGCCGATCCAGTA |
| Sequence-based reagent | Kit | IDT DNA | Mm.PT.58.33701407 | Fwd: CGGCTAACAAAGGGAAGGAT Rev: GTATAAGTGCCTCCTTCTGTGC |
| Sequence-based reagent | Sell | IDT DNA | Mm.PT.58.13849728 | Fwd: CTTCATTCCTGTAGCCGTCAT Rev: CCATCCTTTCTTGAGATTTCTTGC |
| Sequence-based reagent | CD69 | IDT DNA | Mm.PT.58.32284621 | Fwd: ACGGAAAATAGCTCTTCACATCT Rev: ACCACTATTAACACAGCCCAAG |
| Sequence-based reagent | Klrb1b | IDT DNA | Mm.PT.58.41786742 | Fwd: CTAGCCAGGATCCAAGAACC Rev: CAATCACGACCAGCACAAGA |
| Sequence-based reagent | Bhlhe41 | IDT DNA | Mm.PT.31539347 | Fwd: GGAACATAGGGATTTTATAGGACTGG Rev: GCATTCATTAATTCGGTCTCGTC |
| Sequence-based reagent | Sox6 | IDT DNA | Mm.PT.58.30547031 | Fwd: CCGTACAGTTCATTCCGTCAA Rev: GTCACTTATGCCCTTTAGCCT |
| Sequence-based reagent | Ccr7 | IDT DNA | Mm.PT.58.312575202 | Fwd: GAGACAAGAACCAAAAGCACAG Rev: GGAAAATGACAAGGAGAGCCA |
| Sequence-based reagent | CD39 | IDT DNA | Mm.PT.58.8557349 | Fwd: CGAGAAGGAGAATGACACAGG Rev: GTATCAGTTCGGTGGACAGTTC |
| Sequence-based reagent | Irf8 | IDT DNA | Mm.PT.58.30819027 | Fwd: TGTCTCCCTCTTTAAACTTCCC Rev: GAAGACCATGTTCCGTATCCC |
| Sequence-based reagent | Ccl5 | IDT DNA | Mm.PT.58.43548565 | Fwd: GCTCCAATCTTGCAGTCGT Rev: CCTCTATCCTAGCTCATCTCCA |
| Sequence-based reagent | Cx3cr1 | IDT DNA | Mm.PT.58.17555544 | Fwd: TCCCTTCCCATCTGCTCA Rev: CACAATGTCGCCCAAATAACAG |
| Sequence-based reagent | Prdm1 | IDT DNA | Mm.PT.58.10253822 | Fwd: GAACCTGCTTTTCAAGTATGCTG Rev: TTCCCTTCGGTATGTACTCCT |
| Sequence-based reagent | Actb | IDT DNA | Mm.PT. 39a.22214843.g | Fwd: GATTACTGCTCTGGCTCCTAG Rev: GACTCATCGTACTCCTGCTTG |
| Sequence-based reagent | Bach2 Exon1-2 | IDT DNA | Mm.PT.58.12332872 | Fwd: TGTAGCCTTCTCATCTCTTCCT Rev: ATCCACAGACATGCCGTTC |
| Sequence-based reagent | Bach2 Exon3-4 | IDT DNA | Mm.PT.58.11332004 | Fwd: GAAGCAGACAGTGAGTCGTG Rev: ACTGTTCTGAGGTTAGCTTGTG |
| Sequence-based reagent | Bach2 Exon4-5 | IDT DNA | Mm.PT.58.11289601 | Fwd: GAGTTCATCCACGACATCCG Rev: CAGTTTTTCCTTCTCGCACAC |
| Chemical compound, drug | DNase | Sigma | Catalog# DN25 | |

*Appendix 1 Continued on next page*

*Appendix 1 Continued*

| Reagent type (species) or resource | Designation | Source or reference | Identifiers | Additional information |
|---|---|---|---|---|
| Chemical compound, drug | Hyaluronidase | Sigma | Catalog# H2126 | |
| Chemical compound, drug | Liberase | Roche | Catalog# 05401119001 | |
| Peptide, recombinant protein | Recombinant murine IL-15 | PeproTech | Catalog# 210–15 | |
| Peptide, recombinant protein | Recombinant murine IL-12 p70 | PeproTech | Catalog# 200–12 | |
| Peptide, recombinant protein | Recombinant mouse IL-18 | MBL | Catalog# B002-5 | |
| Chemical compound, drug | Calcein-AM | Biolegend | Catalog# 425201 | |
| Software, algorithm | Prism 9 | GraphPad | https://www.graphpad.com/ | |
| Software, algorithm | FlowJo 10 | Treestar | https://www.flowjo.com/ | |

