## [Editor Report]

This paper, by identifying the novel factor BACH2s involvement in the generation and maintenance of NK cells, helps to fully define the NK cell network and will contribute to our understanding or regulation of NK cell development and function.

---

## [Decision Letter]

**Decision letter after peer review:**

Thank you for submitting your article "The Transcription Factor Bach2 Negatively Regulates Natural Killer Cell Maturation and Function" for consideration by *eLife*. Your article has been reviewed by 3 peer reviewers, one of whom is a member of our Board of Reviewing Editors, and the evaluation has been overseen by Satyajit Rath as the Senior Editor. The reviewers have opted to remain anonymous.

Essential revisions:

1. Strengthen the information around the reproducibility and robustness of experimental design, replication and validation and expand the functional assessment of the phenotype.

2. Provide a deeper analysis of the mechanistic and bioinformatic data.

3. Strengthen the analyses of the tumour model including strengthening controls and deeper analyses of NK cell phenotypes.

Overall, revision of this work could significantly improve the quality of the study for the community.

*Reviewer #1 (Recommendations for the authors):*

Specific points that should be addressed:

1. Figure 1F: With the reduction of Bach2 expression between 2 and 12-week old mice, this seems to be correlated with Bach2 expression within the populations – this data has not been shown. Is it possible to show this data in the supplementary Figure 1?

2. Does this switch imply that reduction in Bach2 expression is necessary to make mature NK cells?

3. Did the authors analyse the conditional mice for deletion efficiency? Could these data be provided?

4. NcrCreBach2 mice were generated. The floxed mice were used as controls. It is unclear if NcrCre mice were also used as controls as their expression of NKp46 is diminished compared with wild-type mice, and thus likely Bach2 floxed mice. If NcrCre mice were not used, could the authors provide an explanation for this and indicate in the paper the implications (although it is appreciated that with corrected gating, this should not be an issue).

5. Although the repertoire was not drastically affected, CD94 was altered 'considerably' but was not ablated.

6. The authors provide a nice analysis of the deficient and wildtype cells but have not undertaken extensive confirmation of changes nor explored the molecular relationships or interactions that are driving the changes they see. The work would be much stronger with a much deeper analysis of these features.

*Reviewer #2 (Recommendations for the authors):*

Data presentation could be improved. First of all, all bar graphs should be replaced by scatter plots as in Figure 1G so that the biological replicates and variation could be clearly shown. In Figure 1, there should be statistics accompanying FACS plots. In Figure 3, there is no information on the group size of each experiment so it is difficult to judge the conclusion. For example, Figure 3A shows no significant difference in subsets in BM but the FACS plots suggest there was some difference. Would it be an issue of small group size? If so, larger group sizes (n>5) are preferred.

In the tumor model, more characterization of NK cells in the lung could be helpful. Were there any changes in numbers for total NK cells or specific subsets between WT and KO NK cells? What were the levels of effector functional changes of KO NK cells, as compared to WT NK cells?

*Reviewer #3 (Recommendations for the authors):*

The manuscript raises the following concerns requiring significant changes:

(i) NK cell number is not changed in the spleen of KO mice, but what about other sites – particularly BM, lymph node, liver and lung? Are the subset changes also observed in these peripheral sites? Characterisation of the lung is particularly important as differences in lung homing could explain the augmented lung metastasis control seen in KO mice.

(ii) Related to the above point, does NK cell BACH2 expression change in key peripheral tissue sites (eg. lungs, LN, liver)?

(iii) There is no assessment of NK cell function in KO mice. Do KO NK cells exhibit superior ex vivo killing capacity and cytokine production? This is critical for understanding why KO mice better control tumours.

(iv) The RNAseq data suggests that KO NK cells are more proliferative. Are there more Ki67+ cells and/or is there more EdU incorporation within KO NK cells? This could again be relevant to the proliferative response to tumour challenge.

(v) The phenotype of KO NK cells within the lung early during the response to tumour challenge also needs to be examined to clarify whether there are any KO-specific changes in NK cell number and/or phenotype stimulated by lung metastases.

(vi) Mechanistic data and bioinformatic analysis are minimal. Ideally, WT vs KO ATAC-seq data should be included as BACH2 is known to influence the binding of other key TFs (eg. RUNX3, BATF) via changes in accessibility. However, even cross-referencing the RNAseq data with published data from CD8s, a cell type that the authors propose BACH2 functions similarly within, would add significant value. For example, how many of the DE genes are known as direct Bach2 targets within CD8s (using data from Roychoudhuri et al., NI 2016)? How many of these targets were also identified as bound by Jun factors in the same paper (ie. could AP-1 antagonism explain some of the gene expression changes)?

(vii) In general, a more in-depth discussion of how published BACH2 regulated pathways (ie. BATF, Blimp1, RUNX3, AP-1, etc) could explain the KO phenotype is needed. As noted above, despite superficial similarities between the NK cell and CD8 KO phenotype, the lack of difference in BATF and Blimp1 expression within KO NK cells argues that there may be significant mechanistic differences in how BACH2 operates within these two cell types.

(viii) Given the importance of the tumour control data to the paper, more mice are needed in the Ctrl group in Figure 4B, as two significant outliers appear to be skewing/exaggerating the Ctrl average. Also, 3 mice per group in the PK136 experiments is not enough for publication – at least 5-6 mice per group are needed.

---

## [Author Response]

Reviewer #1 (Recommendations for the authors):Specific points that should be addressed:1. Figure 1F: With the reduction of Bach2 expression between 2 and 12-week old mice, this seems to be correlated with Bach2 expression within the populations – this data has not been shown. Is it possible to show this data in the supplementary Figure 1?

We agree. Accordingly, we have directly compared the expression of Bach2 (by fold change of MFI) within populations of NK cells between 2-week and 12-week-old mice. In Figure 1—figure supplement 1D, we showed that, in 2-week-old mice, Bach2 has significantly higher expression within each subset compared to 12-week-old mice. Changes are described on lines 155-157.

2. Does this switch imply that reduction in Bach2 expression is necessary to make mature NK cells?

We think the reduction of Bach2’s expression might be necessary to make mature NK cells. On the one hand, there are negative correlations between the expression of Bach2 with age and maturation status (Figure 1D and 1F). Also, Bach2 in NK cells functions to repress the expression of effector genes (Figure 2D), which are critical for maturation of NK cells.

3. Did the authors analyse the conditional mice for deletion efficiency? Could these data be provided?

Yes, we have analyzed the deletion efficiency in conditional mice by qPCR and added on lines 170-172. In Figure 2—figure supplement 1A, we detected *Bach2* mRNA expression by 3 pairs of primers. Primers targeting exon 1-2 which includes part of deleted region give no signal while primers targeting exon 3-4 and exon 4-5 as internal controls show normal expression levels. Since the start codon is within deletion region, there will be no translation of Bach2 protein.

4. NcrCreBach2 mice were generated. The floxed mice were used as controls. It is unclear if NcrCre mice were also used as controls as their expression of NKp46 is diminished compared with wild-type mice, and thus likely Bach2 floxed mice. If NcrCre mice were not used, could the authors provide an explanation for this and indicate in the paper the implications (although it is appreciated that with corrected gating, this should not be an issue).

Thank you for your kind reminder. As you have pointed out, we dealt this situation with proper gating. In Figure 2—figure supplement 1C, we showed that there was slightly lower NKp46 expression in Bach2^cKO^ (*Bach2*^flox/flox^
*NKp46*-iCre) mice compared to control (*Bach2*^flox/flox^) mice. However, with proper gating, all the NKp46 positive cells are included.

Additionally, a previous study (Narni-Mancinelli, E., Chaix, J., Fenis, A., Kerdiles, Y. M., Yessaad, N., Reynders, A., Gregoire, C., Luche, H., Ugolini, S., Tomasello, E., Walzer, T., and Vivier, E. (2011). Fate mapping analysis of lymphoid cells expressing the NKp46 cell surface receptor. Proceedings of the National Academy of Sciences of the United States of America, 108(45), 18324–18329. https://doi.org/10.1073/pnas.1112064108) showed that Ncr-Cre does not affect the maturation of NK cells.

5. Although the repertoire was not drastically affected, CD94 was altered 'considerably' but was not ablated.

Thank you for pointing this out. The reduction of CD94 may be irrelevant to the development and function of NK cells based on a previous study (Orr, M. T., Wu, J., Fang, M., Sigal, L. J., Spee, P., Egebjerg, T., Dissen, E., Fossum, S., Phillips, J. H., and Lanier, L. L. (2010). Development and function of CD94-deficient natural killer cells. PloS one, 5(12), e15184. https://doi.org/10.1371/journal.pone.0015184). It showed that the deficiency of CD94 has no role on NK maturation and many functions. Therefore, we did not specifically explore this within the scope of our current study. In the revised paper, we discussed this point on lines 172-177.

6. The authors provide a nice analysis of the deficient and wildtype cells but have not undertaken extensive confirmation of changes nor explored the molecular relationships or interactions that are driving the changes they see. The work would be much stronger with a much deeper analysis of these features.

We have taken this comment to heart by performing additional experiments and have revised the manuscript accordingly.

In Figure 2, we performed qPCR (Figure 2C and 2D) and flow cytometry (Figure 2E and 2F) to confirm the differential expression of target genes. Also, we included more biological replicates to make it more reliable.

In Figure 3—figure supplement 1C and 1D, we extracted all transcription factors that have differential expression with Bach2 deficiency from RNA-seq data and explored the potential interactions between them and found potential targets of Bach2. Changes are described on lines 274-283.

In Figure 3, we performed ATAC-seq to compare the differential accessible regions (DARs) between Bach2^cKO^ mice and control mice. With ATAC-seq data, we checked if there are DARs around the genes for potential Bach2 target transcription factors and if there is a Bach2 binding motif within the DARs. Also, we used flow cytometry to validate the expression of potential Bach2 targets. Changes are described on lines 283-299.

In Figure 3—figure supplement 1A and 1B, we compared our RNA-seq data with published data on Bach2 from CD8^+^ T cells (Roychoudhuri, R., Clever, D., Li, P., Wakabayashi, Y., Quinn, K. M., Klebanoff, C. A., Ji, Y., Sukumar, M., Eil, R. L., Yu, Z., Spolski, R., Palmer, D. C., Pan, J. H., Patel, S. J., Macallan, D. C., Fabozzi, G., Shih, H. Y., Kanno, Y., Muto, A., Zhu, J., … Restifo, N. P. (2016). BACH2 regulates CD8(+) T cell differentiation by controlling access of AP-1 factors to enhancers. Nature immunology, 17(7), 851–860. https://doi.org/10.1038/ni.3441). We pulled out the differentially expressed genes that have been identified as direct Bach2 targets (TCR-induced and non-induced) in CD8 cells and compared the differences between average JunD binding at these Bach2 binding sites with differences in mRNA expression. Changes were added to lines 255-261.

Reviewer #2 (Recommendations for the authors):Data presentation could be improved. First of all, all bar graphs should be replaced by scatter plots as in Figure 1G so that the biological replicates and variation could be clearly shown.

Thank you for your suggestion. We have made revisions accordingly. All bar graphs have been replaced by scatter plots.

In Figure 1, there should be statistics accompanying FACS plots.

The statistics of gMFI have been added to the FACS plots. See changes in Figure 1.

In Figure 3, there is no information on the group size of each experiment so it is difficult to judge the conclusion. For example, Figure 3A shows no significant difference in subsets in BM but the FACS plots suggest there was some difference. Would it be an issue of small group size? If so, larger group sizes (n>5) are preferred.

Thank you for pointing this out. In addition, we have repeated the experiments to add more biological replicates and revised the figures to show the group size (n = 8 for each group from three independent). The revised figure is now referenced as Figure 4A.

In the tumor model, more characterization of NK cells in the lung could be helpful. Were there any changes in numbers for total NK cells or specific subsets between WT and KO NK cells? What were the levels of effector functional changes of KO NK cells, as compared to WT NK cells?

We performed more experiments according to your suggestions.

In Figure 4J, we showed that there are significantly higher numbers for total NK cells in lung in Bach2^cKO^ mice. In Figure 4I, we found there is a higher percentage of CD27-CD11b+ populations in the lung in Bach2^cKO^ mice. Changes are described on lines 326-331.

In Figure 5—figure supplement 1A, we showed that Bach2 deficient NK cells from spleen show similar cytotoxicity towards B16F10 cells compare to Bach2 sufficient NK cells. In Figure 5—figure supplement 1B, we found that there is a similar percentage of IFNγ-producing NK cells comparing Bach2-deficient NK cells and control NK cells upon cytokine stimulation. Changes are described on lines 347-352.

In Figure 5—figure supplement 1C, we showed that Bach2-deficient NK cells in the lung show similar GzmB expression at steady state. Changes are described on lines 352-355.

Reviewer #3 (Recommendations for the authors):The manuscript raises the following concerns requiring significant changes:(i) NK cell number is not changed in the spleen of KO mice, but what about other sites – particularly BM, lymph node, liver and lung? Are the subset changes also observed in these peripheral sites? Characterisation of the lung is particularly important as differences in lung homing could explain the augmented lung metastasis control seen in KO mice.

Thank you for your suggestions. We performed additional experiments to address these questions.

In Figure 4, we show that in BM and lymph node, NK cell numbers are significantly decreased in Bach2 knockout mice (Figure 4B and 4F). In liver, there are similar number of NK cells between Bach2^cKO^ mice and control mice (Figure 4H). Interestingly, we found higher number of NK cells in the lung in Bach2^cKO^ mice (Figure 4J).

In Figure 4E, we showed that in lymph node, there are no subset changes observed. In Figure 4G, we found that in liver, NK cells show similar phenotype to cells from spleen. There are higher percentage of CD27-CD11b+ subsets and lower percentage of DP subsets. In Figure 4I, we noticed that majority of the NK cells in lung are CD27-CD11b+. Bach2 knockout mice showed a higher percentage of this population in lung compared to control mice.

The changes mentioned above are described on lines 305-307 and 322-331.

(ii) Related to the above point, does NK cell BACH2 expression change in key peripheral tissue sites (eg. lungs, LN, liver)?

In Figure 1G, when we compared Bach2 expression in spleen NK cells, we observed similar Bach2 levels in lymph node and liver NK cells. In lung NK cells, Bach2 expression is much lower which is likely because the majority of the NK cells in lung are in the CD27-CD11b+ population, which has the lowest Bach2 expression. The data are described on lines 157-162.

(iii) There is no assessment of NK cell function in KO mice. Do KO NK cells exhibit superior ex vivo killing capacity and cytokine production? This is critical for understanding why KO mice better control tumours.

Thank you for pointing this out. In the revised manuscript, we have included functional assessment data.

In Figure 5—figure supplement 1A, we showed Bach2-deficient NK cells show similar ex vivo killing against B16F10 cells compared to WT NK cells. In Figure 5—figure supplement 1B, we showed that there is a similar percentage of IFNγ-producing NK cells when comparing Bach2-deficient NK cells and control NK cells upon cytokine stimulation. Changes are described on lines 347-352.

(iv) The RNAseq data suggests that KO NK cells are more proliferative. Are there more Ki67+ cells and/or is there more EdU incorporation within KO NK cells? This could again be relevant to the proliferative response to tumour challenge.

Thank you for the suggestion. In Figure 2—figure supplement 1E, we showed at steady state, there is a similar percentage of Ki67+ cells in Bach2 deficient NK cells and WT NK cells (on lines 213-216). In Figure 5—figure supplement 1D, we showed that after 24h of B16F10 tumour challenge, the percentage of Ki67+ cells are comparable between Bach2-deficient and WT NK cells (on lines 359-361).

(v) The phenotype of KO NK cells within the lung early during the response to tumour challenge also needs to be examined to clarify whether there are any KO-specific changes in NK cell number and/or phenotype stimulated by lung metastases.

Thank you for your suggestions. We have added the data to Figure 5.

In Figure 5C to 5E, we showed that there is a higher number of total NK cells, higher percentage of CD27-CD11b+ population and higher percentage of KLRG1+ NK cells in Bach2 knockout mice after 24h of tumor challenge, suggesting the higher percentage of mature NK cells (CD11b+) with elevated KLRG1 expression may contribute to lower metastases colonies in Bach2^cKO^ mice. We discussed this point on lines 355-368.

(vi) Mechanistic data and bioinformatic analysis are minimal. Ideally, WT vs KO ATAC-seq data should be included as BACH2 is known to influence the binding of other key TFs (eg. RUNX3, BATF) via changes in accessibility. However, even cross-referencing the RNAseq data with published data from CD8s, a cell type that the authors propose BACH2 functions similarly within, would add significant value. For example, how many of the DE genes are known as direct Bach2 targets within CD8s (using data from Roychoudhuri et al., NI 2016)? How many of these targets were also identified as bound by Jun factors in the same paper (ie. could AP-1 antagonism explain some of the gene expression changes)?

Thank you for your suggestions. We performed an ATAC-seq experiment to address this point, now shown in Figure 3. The data showed that AP-1 factor binding motif was enriched in upregulated DARs.

Also, we cross-referenced our RNA-seq data to the public data from CD8s (Roychoudhuri, R., Clever, D., Li, P., Wakabayashi, Y., Quinn, K. M., Klebanoff, C. A., Ji, Y., Sukumar, M., Eil, R. L., Yu, Z., Spolski, R., Palmer, D. C., Pan, J. H., Patel, S. J., Macallan, D. C., Fabozzi, G., Shih, H. Y., Kanno, Y., Muto, A., Zhu, J., … Restifo, N. P. (2016). BACH2 regulates CD8(+) T cell differentiation by controlling access of AP-1 factors to enhancers. Nature immunology, 17(7), 851–860. https://doi.org/10.1038/ni.3441). The data are added to Figure 3—figure supplement 1A and 1B in which we compared our RNA-seq data with RNA-seq data from CD8^+^ T cells.

There were 193 differentially expressed genes identified to be direct target of Bach2 in CD8 cells (including 24 TCR-induced genes and 169 non TCR-induced genes). We showed the data of TCR-induced genes in Figure 3—figure supplement 1A. We also compared the differences between average JunD binding at these Bach2 binding sites with differences in mRNA expression in Figure 3—figure supplement 1B. Our data showed that there was indeed increased JunD occupancy in the absence of Bach2 in NK cells.

The data are described in detail on lines 236-269.

(vii) In general, a more in-depth discussion of how published BACH2 regulated pathways (ie. BATF, Blimp1, RUNX3, AP-1, etc) could explain the KO phenotype is needed. As noted above, despite superficial similarities between the NK cell and CD8 KO phenotype, the lack of difference in BATF and Blimp1 expression within KO NK cells argues that there may be significant mechanistic differences in how BACH2 operates within these two cell types.

Thank you for pointing this out. Accordingly, we have done more in-depth analyses of our RNA-seq data.

In Figure 3—figure supplement 1C, we extracted all transcription factors that have differential expression with Bach2 deficiency from RNA-seq data and explored the potential interactions between them and found potential targets of Bach2. For example, *Prdm1* turns out to be significantly upregulated however with only a slight fold-change in Bach2-deficient NK cells. Besides, we have found that *Irf8* is significantly upregulated in Bach2-deficient NK cells. In Figure 3—figure supplement 1D, we explored the interaction between these differentially expressed transcription factors in STRING database and found *Prdm1*, *Irf8* to be potential targets of Bach2. In Figure 3D and 3E, we have used flow cytometry to validate the protein expression change of Blimp-1 and IRF8. In Figure 3, we performed ATAC-seq to compare the differential accessible regions (DARs) between Bach2^cKO^ mice and control mice. With ATAC-seq data, we checked if there are DARs around the genes *Prdm1* and *Irf8* as potential Bach2 target transcription factors and if there are Bach2 binding motifs within the DARs. Changes in detail are added on lines 271-299.

On lines 433-443, we also discussed that with regard to RUNX3 and TCF1, there might be different mechanisms of how Bach2 regulates cell development and differentiation between CD8^+^ T cells and NK cells.

(viii) Given the importance of the tumour control data to the paper, more mice are needed in the Ctrl group in Figure 4B, as two significant outliers appear to be skewing/exaggerating the Ctrl average. Also, 3 mice per group in the PK136 experiments is not enough for publication – at least 5-6 mice per group are needed.

We have repeated the experiment and added more biological replicates as revised in Figure 5B.